# Sufficient conditions for offline reactivation in recurrent neural networks

**Nanda H Krishna**[1,2,✉]   **Colin Bredenberg**[1,2,✉]   **Daniel Levenstein**[1,3]
**Blake Aaron Richards**[1,3,4,5]   **Guillaume Lajoie**[1,2,4,✉]

[1]Mila – Quebec AI Institute   [2]Université de Montréal   [3]McGill University
[4]Canada CIFAR AI Chair   [5]CIFAR Learning in Machines & Brains
✉{nanda.harishankar-krishna,colin.bredenberg,guillaume.lajoie}@mila.quebec

## Abstract

During periods of quiescence, such as sleep, neural activity in many brain circuits resembles that observed during periods of task engagement. However, the precise conditions under which task-optimized networks can autonomously reactivate the same network states responsible for online behavior is poorly understood. In this study, we develop a mathematical framework that outlines sufficient conditions for the emergence of neural reactivation in circuits that encode features of smoothly varying stimuli. We demonstrate mathematically that noisy recurrent networks optimized to track environmental state variables using change-based sensory information naturally develop denoising dynamics, which, in the absence of input, cause the network to revisit state configurations observed during periods of online activity. We validate our findings using numerical experiments on two canonical neuroscience tasks: spatial position estimation based on self-motion cues, and head direction estimation based on angular velocity cues. Overall, our work provides theoretical support for modeling offline reactivation as an emergent consequence of task optimization in noisy neural circuits.

## 1 Introduction

It has been widely observed that neural circuits in the brain recapitulate task-like activity during periods of quiescence, such as sleep (Tingley & Peyrache, 2020). For example, the hippocampus "replays" sequences of represented spatial locations akin to behavioral trajectories during wakefulness (Nádasdy et al., 1999; Lee & Wilson, 2002; Foster, 2017). Furthermore, prefrontal (Euston et al., 2007; Peyrache et al., 2009), sensory (Kenet et al., 2003; Xu et al., 2012), and motor (Hoffman & McNaughton, 2002) cortices reactivate representations associated with recent experiences; and sleep activity in the anterior thalamus (Peyrache et al., 2015) and entorhinal cortex (Gardner et al., 2022) is constrained to the same neural manifolds that represent head direction and spatial position in those circuits during wakefulness.

This neural reactivation phenomenon is thought to have a number of functional benefits, including the formation of long term memories (Buzsáki, 1989; McClelland et al., 1995), abstraction of general rules or "schema" (Lewis & Durrant, 2011), and offline planning of future actions (Ólafsdóttir et al., 2015; Igata et al., 2021). Similarly, replay in artificial systems has been shown to be valuable in reinforcement learning, when training is sparse or expensive (Mnih et al., 2015), and in supervised learning, to prevent catastrophic forgetting in continual learning tasks (Hayes et al., 2019). However, where machine learning approaches tend to save sensory inputs from individual experiences in an external memory buffer, or use external networks that are explicitly trained to generate artificial training data (Hayes & Kanan, 2022), reactivation in the brain is autonomously generated in the same circuits that operate during active perception and action. Currently, it is unknown how reactivation can emerge in the same networks that encode information during active behavior, or why it is so widespread in neural circuits.

Previous approaches to modeling reactivation in neural circuits fall into two broad categories: generative models that have been explicitly trained to reproduce realistic sensory inputs (Deperrois et al.,

2022), and models in which replay is an emergent consequence of the architecture of network models with a particular connectivity structure (Shen & McNaughton, 1996; Azizi et al., 2013; Milstein et al., 2023) or local synaptic plasticity mechanism (Hopfield, 2010; Litwin-Kumar & Doiron, 2014; Theodoni et al., 2018; Haga & Fukai, 2018; Asabuki & Fukai, 2023). Other approaches have modeled replay and theta sequences as emergent consequences of firing rate adaptation (Chu et al., 2024) or input modulation (Kang & DeWeese, 2019) in continuous attractor network models. Generative modeling approaches have strong theoretical guarantees that reactivation will occur, because networks are explicitly optimized to provide this functionality. However, modeling approaches that argue for *emergent* reactivation typically rely on empirical results, and lack rigorous mathematical justification.

In this study, we demonstrate that a certain type of reactivation—diffusive reactivation—can emerge from a system attempting to optimally encode features of its environment in the presence of internal noise. We trained continuous-time recurrent neural networks (RNNs) to optimally integrate and track perceptual variables based on sensations of change (motion through space, angular velocity, etc.), in the context of two ethologically relevant tasks: spatial navigation and head direction integration. Critically, training in this manner has been shown to produce grid cell (Cueva & Wei, 2018; Sorscher et al., 2019) and head direction cell representations (Cueva et al., 2020; Uria et al., 2022), which correspond to neural systems in which reactivation phenomena have been observed (Gardner et al., 2022; Peyrache et al., 2015). We see that these networks exhibit reactivation during quiescent states (when subject to noise but in the absence of perceptual inputs), and we explain these phenomena by demonstrating that noise compensation dynamics naturally induce diffusion on task-relevant neural manifolds in optimally trained networks.

## 2 MATHEMATICAL RESULTS

### 2.1 SETUP AND VARIABLES

In this study, we will consider a noisy discrete-time approximation of a continuous-time RNN, receiving change-based information $\frac{d\mathbf{s}(t)}{dt}$ about an $N_s$-dimensional environmental state vector $\mathbf{s}(t)$. The network's objective will be to reconstruct some function of these environmental state variables, $f(\mathbf{s}(t)) : \mathbb{R}^{N_s} \to \mathbb{R}^{N_o}$, where $N_s$ is the number of stimulus dimensions and $N_o$ is the number of output dimensions (for a schematic, see Fig. 1a). An underlying demand for this family of tasks is that path integration needs to be performed, possibly followed by some computations based on that integration. These requirements are often met in natural settings, as it is widely believed that animals are able to estimate their location in space $\mathbf{s}(t)$ through path integration based exclusively on local motion cues $\frac{d\mathbf{s}(t)}{dt}$, and neural circuits in the brain that perform this computation have been identified (specifically the entorhinal cortex (Sorscher et al., 2019)). For our analysis, we will assume that the stimuli the network receives are drawn from a stationary distribution, such that the probability distribution $p(\mathbf{s}(t))$ does not depend on time—for navigation, this amounts to ignoring the effects of initial conditions on an animal's state occupancy statistics, and assumes that the animal's navigation policy remains constant throughout time. The RNN's dynamics are given by:

$$\mathbf{r}(t + \Delta t) = \mathbf{r}(t) + \Delta \mathbf{r}(t) \tag{1}$$

$$\Delta \mathbf{r}(t) = \phi\left(\mathbf{r}(t), \mathbf{s}(t), \frac{d\mathbf{s}(t)}{dt}\right)\Delta t + \sigma \boldsymbol{\eta}(t), \tag{2}$$

where $\Delta \mathbf{r}(t)$ is a function that describes the network's update dynamics as a function of the stimulus, $\phi(\cdot)$ is a sufficiently expressive nonlinearity, $\boldsymbol{\eta}(t) \sim \mathcal{N}(0, \Delta t)$ is Brownian noise, and $\Delta t$ is taken to be small as to approximate corresponding continuous-time dynamics. We work with a discrete-time approximation here for the sake of simplicity, and also to illustrate how the equations are implemented in practice during simulations. Suppose that the network's output is given by $\mathbf{o} = \mathbf{D}\mathbf{r}(t)$, where $\mathbf{D}$ is an $N_o \times N_r$ matrix that maps neural activity to outputs, and $N_r$ is the number of neurons in the RNN.

We formalize our loss function for each time point as follows:

$$\mathcal{L}(t) = \mathbb{E}_{\boldsymbol{\eta}}\left\| f(\mathbf{s}(t)) - \mathbf{D}\mathbf{r}(t) \right\|_2, \tag{3}$$

so that as the loss is minimized over timesteps, the system is optimized to match its target at every timestep while compensating for its own intrinsic noise. Our analysis proceeds as follows. First, we will derive an upper bound for this loss that partitions the optimal update $\Delta \mathbf{r}$ into two terms:

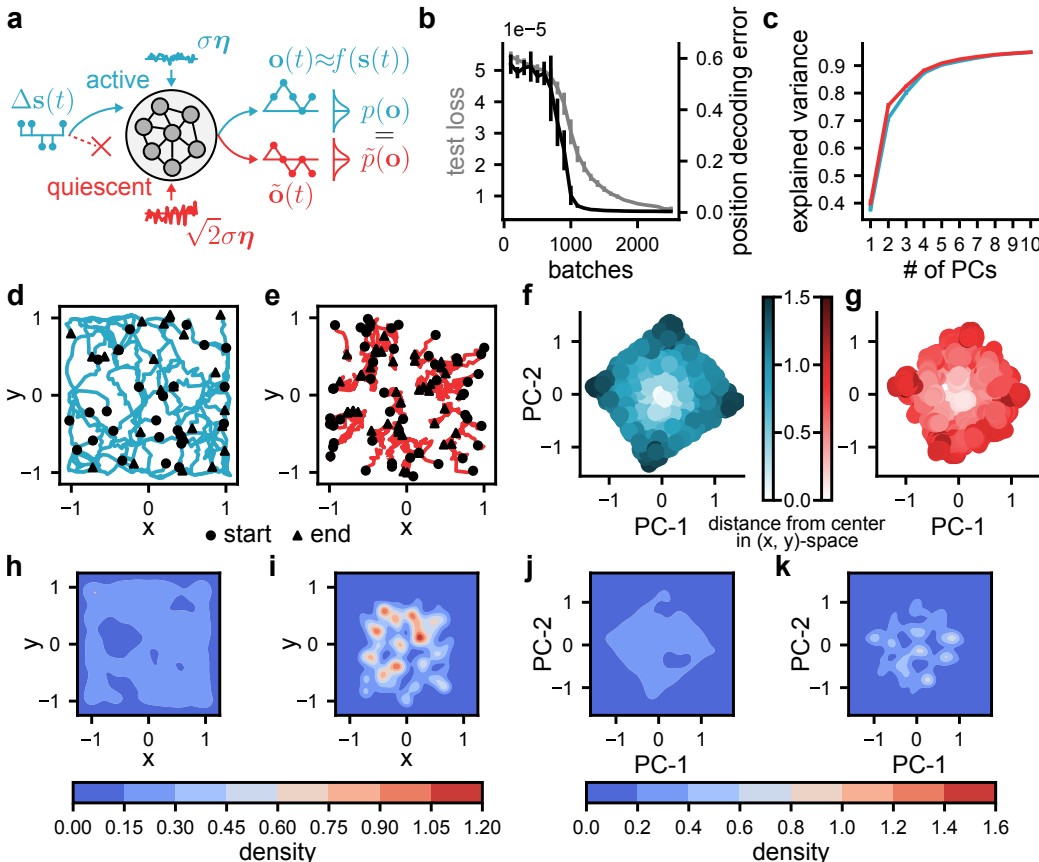

Figure 1: **Reactivation in a spatial position estimation task**. **a)** Schematic of task. **b)** Test metrics as a function of training batches for the place cell tuning mean squared error loss used during training (gray) and for the position decoding spatial distance error (black). **c)** Explained variance as a function of the number of principal components (PCs) in population activity space during task engagement (blue) and during quiescent noise-driven activity (red). **d)** Sample decoded outputs during active behavior. Circles indicate the initial location, triangles indicate the final location. **e)** Same as (d), but for decoded outputs during quiescence. **f)** Neural activity projected onto the first two PCs during the active phase. Color intensity measures the decoded output's distance from the center in space. **g)** Neural activity during the quiescent phase projected onto the same active PC axes as in (f). **h)** Two-dimensional kernel density estimate (KDE) plot measuring the probability of state-occupancy over 200 decoded output trajectories during active behavior. **i-k)** Same as (h), but for decoded outputs during quiescence (i), neural activity projected onto the first two PCs during the active phase (j), and during the quiescent phase (k). Error bars (b-c) indicate $\pm 1$ standard deviation over five networks.

one which requires the RNN to estimate the predicted change in the target function, and one which requires the RNN to compensate for the presence of noise. Second, we derive closed-form optimal dynamics for an upper bound of this loss, which reveals a decomposition of neural dynamics into state estimation and denoising terms. Lastly, we show that under certain conditions, these optimal dynamics can produce offline sampling of states visited during training in the absence of stimuli but in the presence of noise.

## 2.2 UPPER BOUND OF THE LOSS

To derive our upper bound, we first assume that $\phi\left(\mathbf{r}(t), \mathbf{s}(t), \frac{\mathrm{d}\mathbf{s}(t)}{\mathrm{d}t}\right)$ can be decomposed into two different functions so that Eq. 2 becomes:

$$\Delta\mathbf{r}(t) = \Delta\mathbf{r}_1\big(\mathbf{r}(t)\big) + \Delta\mathbf{r}_2\left(\mathbf{r}(t), \mathbf{s}(t), \frac{\mathrm{d}\mathbf{s}(t)}{\mathrm{d}t}\right) + \sigma\boldsymbol{\eta}(t), \tag{4}$$

where we will ultimately show that both functions scale with $\Delta t$. We are assuming that these two terms have different functional dependencies; however, for notational conciseness, we will subsequently refer to both updates as $\Delta \mathbf{r}_1(t)$ and $\Delta \mathbf{r}_2(t)$. The first, $\Delta \mathbf{r}_1(t)$, is a function of $\mathbf{r}(t)$ only, and will be used to denoise $\mathbf{r}(t)$ such that the approximate equality $\mathbf{r}(t) + \Delta \mathbf{r}_1(t) + \sigma \boldsymbol{\eta}(t) \approx \mathbf{D}^{\dagger} f(\mathbf{s}(t))$ still holds ($\Delta \mathbf{r}_1(t)$ cancels out the noise corruption $\sigma \boldsymbol{\eta}(t)$), where $\mathbf{D}^{\dagger}$ is the right pseudoinverse of $\mathbf{D}$. This maintains optimality in the presence of noise. The second, $\Delta \mathbf{r}_2(t)$, is also a function of the input, and will build upon the first update to derive a simple state update such that $\mathbf{D}(\mathbf{r}(t+\Delta t)) \approx f(\mathbf{s}(t+\Delta t))$. To construct this two-step solution, we first consider an upper bound on our original loss, which we will label $\mathcal{L}_{upper}$. Exploiting the fact that $\Delta t^2$ is infinitesimally small relative to other terms, we will Taylor expand Eq. 3 to first order about a small timestep increment $\Delta t$:

$$\mathcal{L}(t + \Delta t) = \mathbb{E}_{\boldsymbol{\eta}} \| f(\mathbf{s}(t + \Delta t)) - \mathbf{D}(\mathbf{r}(t) + \Delta \mathbf{r}(t)) \|_2 \tag{5}$$

$$\approx \mathbb{E}_{\boldsymbol{\eta}} \left\| f(\mathbf{s}(t)) + \frac{\mathrm{d}f(\mathbf{s}(t))}{\mathrm{d}\mathbf{s}(t)} \Delta \mathbf{s}(t) - \mathbf{D}(\mathbf{r}(t) + \Delta \mathbf{r}(t)) \right\|_2 \tag{6}$$

$$= \mathbb{E}_{\boldsymbol{\eta}} \left\| \frac{\mathrm{d}f(\mathbf{s}(t))}{\mathrm{d}\mathbf{s}(t)} \Delta \mathbf{s}(t) - \mathbf{D}\Delta \mathbf{r}_2(t) + f(\mathbf{s}(t)) - \mathbf{D}(\mathbf{r}(t) + \Delta \mathbf{r}_1(t) + \sigma \boldsymbol{\eta}(t)) \right\|_2, \tag{7}$$

where $\Delta \mathbf{s}(t) = \frac{\mathrm{d}\mathbf{s}(t)}{\mathrm{d}t} \Delta t$. Next, using the triangle inequality, we note that the loss is upper bounded by a new loss $\mathcal{L}_2$, given by:

$$\mathcal{L} \leq \mathcal{L}_2 = \mathbb{E}_{\boldsymbol{\eta}} \left[ \left\| \frac{\mathrm{d}f(\mathbf{s}(t))}{\mathrm{d}\mathbf{s}(t)} \Delta \mathbf{s}(t) - \mathbf{D}\Delta \mathbf{r}_2(t) \right\|_2 + \left\| f(\mathbf{s}(t)) - \mathbf{D}(\mathbf{r}(t) + \Delta \mathbf{r}_1(t) + \sigma \boldsymbol{\eta}(t)) \right\|_2 \right] \tag{8}$$

$$= \left\| \frac{\mathrm{d}f(\mathbf{s}(t))}{\mathrm{d}\mathbf{s}(t)} \Delta \mathbf{s}(t) - \mathbf{D}\Delta \mathbf{r}_2(t) \right\|_2 + \mathbb{E}_{\boldsymbol{\eta}} \| f(\mathbf{s}(t)) - \mathbf{D}(\mathbf{r}(t) + \Delta \mathbf{r}_1(t) + \sigma \boldsymbol{\eta}(t)) \|_2, \tag{9}$$

which separates the loss into two independent terms: one which is a function of $\Delta \mathbf{r}_2(t)$ and the signal, while the other is a function of $\Delta \mathbf{r}_1(t)$ and the noise. The latter, $\Delta \mathbf{r}_1(t)$-dependent term, allows $\Delta \mathbf{r}_1$ to correct for noise-driven deviations between $f(\mathbf{s}(t))$ and $\mathbf{D}\mathbf{r}(t)$. Here, we will assume that this optimization has been successful for previous timesteps, such that $\mathbf{r}(t) \approx \mathbf{D}^{\dagger} f(\mathbf{s}(t))$, where $\mathbf{D}^{\dagger}$ is the right pseudoinverse of $\mathbf{D}$. By this assumption, we have the following approximation:

$$\mathcal{L}_2 \approx \left\| \frac{\mathrm{d}f(\mathbf{s}(t))}{\mathrm{d}\mathbf{s}(t)} \Delta \mathbf{s}(t) - \mathbf{D}\Delta \mathbf{r}_2(t) \right\|_2 + \mathbb{E}_{\boldsymbol{\eta}} \| f(\mathbf{s}(t)) - \mathbf{D}(\mathbf{D}^{\dagger} f(\mathbf{s}(t)) + \Delta \mathbf{r}_1(t) + \sigma \boldsymbol{\eta}(t)) \|_2 \tag{10}$$

$$= \left\| \frac{\mathrm{d}f(\mathbf{s}(t))}{\mathrm{d}\mathbf{s}(t)} \Delta \mathbf{s}(t) - \mathbf{D}\Delta \mathbf{r}_2(t) \right\|_2 + \mathbb{E}_{\boldsymbol{\eta}} \| \mathbf{D}(\Delta \mathbf{r}_1(t) + \sigma \boldsymbol{\eta}(t)) \|_2. \tag{11}$$

Thus, the second term in $\mathcal{L}_2$ trains $\Delta \mathbf{r}_1$ to greedily cancel out noise in the system $\sigma \boldsymbol{\eta}(t)$. To show that this is similar to correcting for deviations between $\mathbf{D}^{\dagger} f(\mathbf{s}(t))$ and $\mathbf{r}$ in neural space (as opposed to output space), we use the Cauchy-Schwarz inequality to develop the following upper bound:

$$\mathcal{L}_2 \leq \mathcal{L}_3 = \left\| \frac{\mathrm{d}f(\mathbf{s}(t))}{\mathrm{d}\mathbf{s}(t)} \Delta \mathbf{s}(t) - \mathbf{D}\Delta \mathbf{r}_2(t) \right\|_2 + \|\mathbf{D}\|_2 \mathbb{E}_{\boldsymbol{\eta}} \| \Delta \mathbf{r}_1(t) + \sigma \boldsymbol{\eta}(t) \|_2. \tag{12}$$

This allows the system to optimize for denoising without having access to its outputs, allowing for computations that are more localized to the circuit. As our final step, we use Jensen's inequality for expectations to derive the final form of our loss upper bound:

$$\mathcal{L}_3 \leq \mathcal{L}_{upper} = \left\| \frac{\mathrm{d}f(\mathbf{s}(t))}{\mathrm{d}\mathbf{s}(t)} \Delta \mathbf{s}(t) - \mathbf{D}\Delta \mathbf{r}_2(t) \right\|_2 + \|\mathbf{D}\|_2 \sqrt{\mathbb{E}_{\boldsymbol{\eta}} \| \Delta \mathbf{r}_1(t) + \sigma \boldsymbol{\eta}(t) \|_2^2} \tag{13}$$

$$= \mathcal{L}_{signal}(\Delta \mathbf{r}_2) + \mathcal{L}_{noise}(\Delta \mathbf{r}_1), \tag{14}$$

where $\mathcal{L}_{signal} = \left\| \frac{\mathrm{d}f(\mathbf{s}(t))}{\mathrm{d}\mathbf{s}(t)} \Delta \mathbf{s}(t) - \mathbf{D}\Delta \mathbf{r}_2(t) \right\|_2$ is dedicated to tracking the state variable $\mathbf{s}(t)$, and $\mathcal{L}_{noise} = \|\mathbf{D}\|_2 \sqrt{\mathbb{E}_{\boldsymbol{\eta}} \| \Delta \mathbf{r}_1(t) + \sigma \boldsymbol{\eta}(t) \|_2^2}$ is dedicated to denoising the network state. In the next section, we will describe how this objective function can be analytically optimized in terms of $\Delta \mathbf{r}_1(t)$ and $\Delta \mathbf{r}_2(t)$ in a way that decomposes the trajectory tracking problem into a combination of state estimation and denoising.

## 2.3 OPTIMIZING THE UPPER BOUND

Our optimization will be greedy, so that for each loss $\mathcal{L}(t+\Delta t)$ we will optimize only $\Delta \mathbf{r}(t)$, ignoring dependencies on updates from previous time steps. $\mathcal{L}_{noise}$ is the only term in our loss that depends on $\Delta \mathbf{r}_1(t)$. Ignoring proportionality constants and the square root (which do not affect the location of minima), we have the following equivalence:

$$\operatorname{argmin}_{\Delta \mathbf{r}_1(t)} \mathcal{L}_{noise} \equiv \operatorname{argmin}_{\Delta \mathbf{r}_1(t)} \mathbb{E}_{\boldsymbol{\eta}} \|\Delta \mathbf{r}_1(t) + \sigma \boldsymbol{\eta}(t)\|_2^2. \tag{15}$$

Essentially, the objective of $\Delta \mathbf{r}_1(t)$ is to cancel out the noise $\sigma \boldsymbol{\eta}(t)$ as efficiently as possible, given access to information about $\mathbf{r}(t)$. This is a standard denoising objective function, where an input signal is corrupted by additive Gaussian noise, with the following well-known solution (Miyasawa, 1961; Raphan & Simoncelli, 2011):

$$\Delta \mathbf{r}_1^*(t) = \sigma^2 \frac{\mathrm{d}}{\mathrm{d}\mathbf{r}(t)} \log p\big(\mathbf{r}(t)\big) \Delta t, \tag{16}$$

where $p(\mathbf{r}) = \int p\big(\mathbf{r}(t)\big|\mathbf{s}(t)\big) p\big(\mathbf{s}(t)\big) \mathrm{d}\mathbf{s}(t)$ is the probability distribution over noisy network states given input stimulus $\mathbf{s}(t)$, prior to the application of state updates $\Delta \mathbf{r}_1(t)$ and $\Delta \mathbf{r}_2(t)$. By assumption, $p\big(\mathbf{r}(t)\big|\mathbf{s}(t)\big) \sim \mathcal{N}\big(\mathbf{D}^\dagger f\big(\mathbf{s}(t)\big), \sigma^2 \Delta t\big)$. We note that $\frac{\mathrm{d}}{\mathrm{d}\mathbf{r}(t)} \log p\big(\mathbf{r}(t)\big)$ is the same function for all time points $t$, because for the stimulus sets we consider, $p\big(\mathbf{s}(t)\big)$ does not depend on time (it is a stationary distribution); this demonstrates that the optimal greedy denoising update is not time-dependent. These dynamics move the network state towards states with higher probability, and do not require explicit access to noise information $\boldsymbol{\eta}(t)$.

Next, we optimize for $\Delta \mathbf{r}_2(t)$. $\mathcal{L}_{signal}$ is the only term in $\mathcal{L}_{upper}$ that depends on $\Delta \mathbf{r}_2(t)$, so we minimize:

$$\mathcal{L}_{signal} = \left\| \frac{\mathrm{d}f\big(\mathbf{s}(t)\big)}{\mathrm{d}\mathbf{s}(t)} \Delta \mathbf{s}(t) - \mathbf{D}\Delta \mathbf{r}_2(t) \right\|_2. \tag{17}$$

By inspection, the optimum is given by: $\Delta \mathbf{r}_2^*(t) = \mathbf{D}^\dagger \frac{\mathrm{d}f(\mathbf{s}(t))}{\mathrm{d}\mathbf{s}(t)} \Delta \mathbf{s}(t)$. Thus the full greedily optimal dynamics, in the presence of noise, are given by:

$$\Delta \mathbf{r}^*(t) = \left[ \sigma^2 \frac{\mathrm{d}}{\mathrm{d}\mathbf{r}(t)} \log p\big(\mathbf{r}(t)\big) + \mathbf{D}^\dagger \frac{\mathrm{d}f\big(\mathbf{s}(t)\big)}{\mathrm{d}\mathbf{s}(t)} \frac{\mathrm{d}\mathbf{s}(t)}{\mathrm{d}t} \right] \Delta t + \sigma \boldsymbol{\eta}(t). \tag{18}$$

This heuristic solution provides interpretability to any system attempting to maintain a relationship to a stimulus in the presence of noise. First, denoise the system ($\Delta \mathbf{r}_1^*$). Second, use instantaneous changes in the state variable ($\frac{\mathrm{d}\mathbf{s}(t)}{\mathrm{d}t}$) to update state information. In theory, this solution should hold for any trained network capable of arbitrarily precise function approximation (Funahashi & Nakamura, 1993). In the next section, we will demonstrate that this approximately optimal solution will have interesting emergent consequences for neural dynamics in the continuous-time limit ($\Delta t \to 0$).

## 2.4 EMERGENT OFFLINE REACTIVATION

Having derived our greedily optimal dynamics, we are in a position to ask: what happens in the absence of any input to the system, as would be observed in a quiescent state? We will make two assumptions for our model of the quiescent state: 1) $\frac{\mathrm{d}\mathbf{s}(t)}{\mathrm{d}t} = 0$, so that no time-varying input is being provided to the system, and 2) the variance of the noise is increased by a factor of two (deviating from this factor is not catastrophic as discussed below). This gives the following quiescent dynamics $\tilde{\mathbf{r}}$:

$$\Delta \tilde{\mathbf{r}}(t) = \left[ \sigma^2 \frac{\mathrm{d}}{\mathrm{d}\mathbf{r}(t)} \log p\big(\mathbf{r}(t)\big) \right] \Delta t + \sqrt{2}\sigma \boldsymbol{\eta}(t). \tag{19}$$

Interestingly, this corresponds to Langevin sampling of $p(\mathbf{r})$ (Besag, 1994). Therefore, we can predict an equivalence between the steady-state quiescent sampling distribution $\tilde{p}(\mathbf{r})$ and the active probability distribution over neural states $p(\mathbf{r})$ (so that $p(\mathbf{r}) = \tilde{p}(\mathbf{r})$, and consequently $p(\mathbf{o}) = \tilde{p}(\mathbf{o})$). There are two key components that made this occur: first, the system needed to be performing near-optimal noisy state estimation; second, the system state needed to be determined purely by integrating changes in sensory variables of interest. The final assumption—that noise is doubled during quiescent states—is necessary only to produce sampling from the *exact* same distribution $p(\mathbf{r})$. Different noise variances will result in sampling from similar steady-state distributions with different temperature parameters. When these conditions are present, we can expect to see statistically faithful reactivation phenomena during quiescence in optimally trained networks.

## 3   NUMERICAL SIMULATION TASK SETUP

To validate our mathematical results, we consider numerical experiments on two canonical neuroscience tasks, both of which conform to the structure of the general estimation task considered in our mathematical analysis (Fig. 1a). For each task, we minimize the mean squared error between the network output $\mathbf{o}$ and the task-specific target given by $f(\mathbf{s}(t))$, summed across timesteps.

**Spatial Position Estimation.**   In this task, the network must learn to path integrate motion cues in order to estimate an animal's spatial location in a 2D environment. We first generate an animal's motion trajectories $\mathbf{s}_{SP}(t)$ using the model described in Erdem & Hasselmo (2012). Next, we simulate the activities of $n_{SP}$ place cells for all positions visited. The simulated place cells' receptive field centers $\mathbf{c}^{(i)}$ (where $i = 1, \ldots, n_{SP}$) are randomly and uniformly scattered across the 2D environment, and the activity of each for a position $\mathbf{s}$ is given by the following Gaussian tuning curve:

$$f_{SP}^{(i)}(\mathbf{s}) = \exp\left(-\frac{\|\mathbf{s} - \mathbf{c}^{(i)}\|_2^2}{2\sigma_{SP}^2}\right), \tag{20}$$

where $\sigma_{SP}$ is the scale. We then train our network to output these simulated place cell activities based on velocity inputs ($\Delta\mathbf{s}_{SP}(t)$) from the simulated trajectories. To estimate the actual position in the environment from the network's outputs, we average the centers associated with the top $k$ most active place cells. Our implementation is consistent with prior work (Banino et al., 2018; Sorscher et al., 2019) and all task hyperparameters are listed in Suppl. Table A.1.

**Head Direction Estimation.**   The network's goal in this task is to estimate an animal's bearing $\mathbf{s}_{HD}(t)$ in space based on angular velocity cues $\Delta\mathbf{s}_{HD}(t)$, where $\mathbf{s}(t)$ is a 1-dimensional circular variable with domain $[-\pi, \pi)$. As in the previous task, we first generate random head rotation trajectories. The initial bearing is sampled from a uniform distribution $\mathcal{U}(-\pi, \pi)$, and random turns are sampled from a normal distribution $\mathcal{N}(0, 11.52)$—this is consistent with the trajectories used in the previous task, but we do not simulate any spatial information. We then simulate the activities of $n_{HD}$ head direction cells whose preferred angles $\theta_i$ (where $i = 1, \ldots, n_{HD}$) are uniformly spaced between $-\pi$ and $\pi$, using an implementation similar to the RatInABox package (George et al., 2024). The activity of the $i^{\text{th}}$ cell for a bearing $\mathbf{s}$ is given by the following von Mises tuning curve:

$$f_{HD}^{(i)}(\mathbf{s}) = \frac{\exp\left(\sigma_{HD}^{-2}\cos\left(\mathbf{s} - \theta^{(i)}\right)\right)}{2\pi I_0\left(\sigma_{HD}^{-2}\right)}, \tag{21}$$

where $\sigma_{HD}$ is the spread parameter for the von Mises distribution. With these simulated trajectories, we train the network to estimate the simulated head direction cell activities using angular velocity as input. We estimate the actual bearing from the network's outputs by taking the circular mean of the top $k$ most active cells' preferred angles. Hyperparameters for this task are listed in Suppl. Table A.2.

**Continuous-time RNNs.**   For our numerical experiments, we use noisy "vanilla" continuous-time RNNs with linear readouts (further details provided in Suppl. Section A.3). The network's activity is transformed by a linear mapping to predicted place cell or head direction cell activities. During the quiescent phase, we simulated network activity in the absence of stimuli, and doubled the noise variance, as prescribed by our mathematical analysis (Section 2.4).

## 4   NUMERICAL RESULTS

**Spatial Position Estimation.**   We used several different measures to compare the distributions of neural activity in our trained networks during the spatial position estimation task in order to validate our mathematical analysis. First, we found that the explained variance curves as a function of ordered principal components (PCs) for both the active and quiescent phases were highly overlapping and indicated that the activity manifold in both phases was low-dimensional (Fig. 1c). It is also clear that decoded output activity during the quiescent phase is smooth, and tiles output space similarly to trajectories sampled during the waking phase (Fig. 1d-e). This trend was recapitulated by quiescent neural activity projected onto the first two PCs calculated during the active phase (Fig. 1f-g). To quantify in more detail the similarity in the *distributions* of activity during the active and quiescent

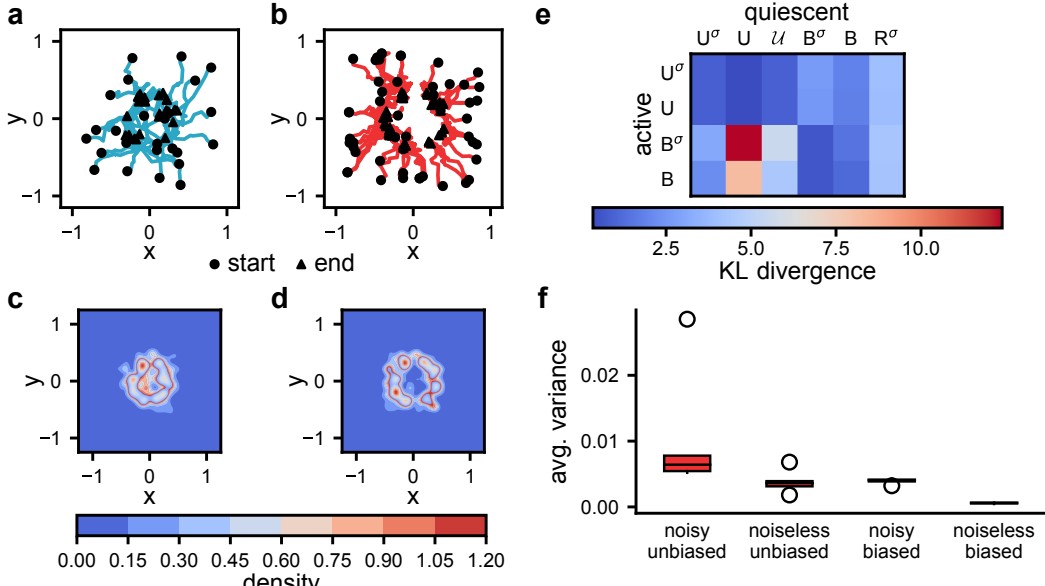

Figure 2: **Biased behavioral sampling and distribution comparisons for spatial position estimation. a-b)** Decoded positions for networks trained under biased behavioral trajectories for the active (a) and quiescent (b) phases. **c-d)** KDE plots for 200 decoded active (c) and quiescent (d) output trajectories. **e)** KL divergence (nats) between KDE estimates for active and quiescent phases. U = unbiased uniform networks, B = biased networks, $\mathcal{U}$ = the true uniform distribution, R = random networks, and the $\sigma$ superscript denotes noisy networks. Values are averaged over five networks. **f)** Box and whisker plots of the total variance (variance summed over output dimensions) of quiescent trajectories, averaged over 500 trajectories. Each plot (e-f) is for five trained networks.

phases, we computed two-dimensional kernel density estimates (KDEs) on the output trajectories (Fig. 1h-i) and on neural activity projected onto the first two active phase PCs (Fig. 1j-k). As our analysis predicts, we indeed found that the distribution of activity was similar across active and quiescent phases, though notably in the quiescent phase output trajectories and neural activities do not tile space as uniformly as was observed during the active phase.

Our theory additionally predicts that if the distribution of network states during the active phase is biased in some way during training, the distribution during the quiescent phase should also be biased accordingly. To test this, we modified the behavioral policy of our agent during training, introducing a drift term that caused it to occupy a ring of spatial locations in the center of the field rather than uniformly tiling space (Fig. 2a). We found again a close correspondence between active and quiescent decoded output trajectories (Fig. 2b), which was also reflected in the KDEs (Fig. 2c-d). These results hold whether or not we increase the variance of noise during quiescence (Suppl. Fig. C.1). We further found that these results hold for continuous-time gated recurrent units (GRUs) (Jordan et al., 2021) (Suppl. Section A.3 and Suppl. Fig. C.2), showing that these reactivation phenomena are not unique to a particular network architecture or activation function. In practice, GRUs were more sensitive to increases in quiescent noise, and other architectures would require more hyperparameter tuning.

To compare activity distributions more quantitatively, we estimated the KL divergence of the distribution of active phase output positions to the distribution of quiescent phase decoded output positions using Monte Carlo approximation (Fig. 2e). We compared outputs from both biased and unbiased distributions, and as baselines, we compared to a true uniform distribution, as well as decoded output trajectories generated by random networks. By our metric, we found that unbiased quiescent outputs were almost as close to unbiased active outputs as a true uniform distribution. Similarly, biased quiescent outputs closely resembled biased active outputs, while biased-to-unbiased, biased-to-random, and unbiased-to-random comparisons all diverged. These results verify that during quiescence, our trained networks do indeed approximately sample from the waking trajectory distribution.

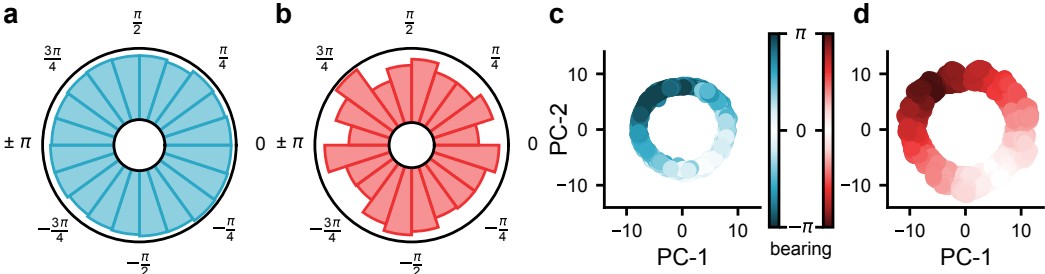

Figure 3: **Reactivation in a head direction estimation task. a-b)** Distribution of decoded head direction bearing angles during the active (a) and quiescent (b) phases. **c-d)** Neural network activity projected onto the first two active phase PCs for active (c) and quiescent (d) phase trajectories. Color bars indicate the decoded output head direction.

We decided to further test the necessity of training and generating quiescent network activity in the presence of noise. By the same KL divergence metric, we found that even trajectories generated by networks that were not trained in the presence of noise, and also were not driven by noise in the quiescent phase, still generated quiescent activity distributions that corresponded well to the active phase distributions. This is likely due to the fact that even networks trained in the absence of noise still learned attractive task manifolds that reflected the agent's trajectory sampling statistics. However, we found that networks without noise in the quiescent state exhibited less variable trajectories, as measured by their steady-state total variance (Fig. 2f). This demonstrates that individual quiescent noiseless trajectories explored a smaller portion of the task manifold than did noisy trajectories (see Suppl. Fig. C.3a-d for a comparison of example noisy and noiseless quiescent trajectories). This failure of exploration could not be resolved by adding additional noise to networks during the quiescent phase: we found that without training in the presence of noise, quiescent phase activity with an equivalent noise level generated erratic, non-smooth decoded output trajectories (Suppl. Fig. C.3e-f), with much higher average distance between consecutive points in a trajectory than quiescent trajectories associated with noisy training (Suppl. Fig. C.4a-b). Thus, noisy training stabilizes noisy quiescent activity, which in turn explores more of the task manifold than noiseless quiescent activity.

**Head Direction Estimation.** To demonstrate the generality of our results across tasks, we also examined the reactivation phenomenon in the context of head direction estimation. Here, as in the previous task, we found that the distribution of decoded head direction bearings closely corresponded across the active and quiescent phases (Fig. 3a-b). Furthermore, we found that the distributions of neural trajectories, projected onto the first two active phase PCs, closely corresponded across both phases (Fig. 3c-d), showing apparent sampling along a ring attractor manifold. To explore whether reactivation dynamics also recapitulate the moment-to-moment transition structure of waking activity, we biased motion in the head direction system to be counter-clockwise (Suppl. Fig. C.5). We found that quiescent trajectories partially recapitulated the transition structure of active phase trajectories, and maintained their bearing for longer periods, resembling real neural activity more closely. The biased velocities were reflected during quiescence, but less clearly than during waking. However, reversals in the trajectories still occurred. These results demonstrate that the type of "diffusive" rehearsal dynamics explored by our theory are still able to produce the temporally correlated, sequential reactivation dynamics observed in the head direction system (Peyrache et al., 2015).

## 5 DISCUSSION

In this study, we have provided mathematical conditions under which reactivation is expected to emerge in task-optimized recurrent neural circuits. Our results come with several key conditions and caveats. Our conditions are as follows: first, the network must implement a noisy, continuous-time dynamical system; second, the network must be solving a state variable estimation task near-optimally, by integrating exclusively change-based inputs ($\frac{d\mathbf{s}(t)}{dt}$) to reconstruct some function of the state variables ($f(\mathbf{s}(t))$) (for a full list of assumptions see Suppl. Section B). Under these conditions, we demonstrated that a greedily optimal solution to the task involves a combination of integrating the

state variables and denoising. In absence of inputs (quiescent phase), we assumed that the system would receive no stimuli ($\frac{\mathrm{d}\mathbf{s}(t)}{\mathrm{d}t} = 0$) so that the system is dominated by its denoising dynamics, and that noise variance would increase slightly (by a factor of 2). Under these conditions, we showed that the steady-state probability distribution of network states during quiescence ($\tilde{p}(\mathbf{r})$) should be equivalent to the distribution of network states during active task performance ($p(\mathbf{r})$). Thus, these conditions constitute criteria for a form of reactivation to emerge in trained neural systems. Note that though we empirically observe reactivation that mimics the moment-to-moment transition structure of waking networks (Suppl. Fig. C.5), our theory does not yet explain how this phenomenon emerges.

We have validated our mathematical results empirically in two tasks with neuroscientific relevance. The first, a path integration task, required the network to identify its location in space based on motion cues. This form of path integration has been used to model the entorhinal cortex (Sorscher et al., 2019), a key brain area in which reactivation dynamics have been observed (Gardner et al., 2019; 2022). The second task required the network to estimate a head direction orientation based on angular velocity cues. This function in the mammalian brain has been attributed to the anterodorsal thalamic nucleus (ADn) and post-subiculum (PoS) (Taube et al., 1990; Taube, 1995; 2007), another critical locus for reactivation dynamics (Peyrache et al., 2015). Previous models have been able to reproduce these reactivation dynamics, by embedding a smooth attractor in the network's recurrent connectivity along which activity may diffuse during quiescence (Burak & Fiete, 2009; Khona & Fiete, 2022). Similarly, we identified attractors in our trained networks' latent activation space—we found a smooth map of space in the spatial navigation task (Fig. 1f-g) and a ring attractor in the head direction task (Fig. 3c-d). In our case, these attractors emerged from task training, and consequently did not require hand crafting—similar to previous work positing that predictive learning could give rise to hippocampal representations (Recanatesi et al., 2021). Furthermore, beyond previous studies, we were able to show that the statistics of reactivation in our trained networks mimicked the statistics of activity during waking behavior, and that manipulation of waking behavioral statistics was directly reflected in offline reactivation dynamics (Fig. 2). Thus, our work complements these previous studies by providing a mathematical justification for the emergence of reactivation dynamics in terms of optimal task performance.

Our results suggest that reactivation in the brain could be a natural consequence of learning in the presence of noise, rather than the product of an explicit generative demand (Hinton et al., 1995; Deperrois et al., 2022). Thus, quiescent reactivation in a brain area should not be taken as evidence for only generative modeling: the alternative, as identified by our work, is that reactivation could be an emergent consequence of task optimization (though it could be used for other computations). Our hypothesis and generative modeling hypotheses may be experimentally dissociable: while generative models necessarily recapitulate the moment-to-moment transition statistics of sensory data, our approach only predicts that the *stationary distribution* will be identical (Section 2.4). This, for instance, opens the possibility for changes in the timescale of reactivation (Nádasdy et al., 1999).

While the experiments explored in this study focus on self-localization and head direction estimation, there are many more systems in which our results may be applicable. In particular, while the early visual system does not require sensory estimation from exclusively change-based information, denoising is a critical aspect of visual computation, having been used for deblurring, occlusion inpainting, and diffusion-based image generation (Kadkhodaie & Simoncelli, 2021)—the mathematical principles used for these applications are deeply related to those used to derive our denoising dynamics. As a consequence, it is possible that with further development our results could also be used to explain similar reactivation dynamics observed in the visual cortex (Kenet et al., 2003; Xu et al., 2012). Furthermore, the task computations involved in head direction estimation are nearly identical to those used in canonical visual working memory tasks in neuroscience (both develop ring attractor structures) (Renart et al., 2003). In addition, evidence integration in decision making involves similar state-variable integration dynamics as used in spatial navigation, where under many conditions the evidence in favor of two opposing decisions is integrated along a line attractor rather than a two-dimensional spatial map (Cain et al., 2013; Mante et al., 2013). Thus our results could potentially be used to model reactivation dynamics observed in areas of the brain dedicated to higher-order cognition and decision making, such as the prefrontal cortex (Peyrache et al., 2009).

In conclusion, our work could function as a justification for a variety of reactivation phenomena observed in the brain. It may further provide a mechanism for inducing reactivation in neural circuits in order to support critical maintenance functions, such as memory consolidation or learning.

## Reproducibility Statement

Our code is available on GitHub at `https://github.com/nandahkrishna/RNNReactivation`. All hyperparameter values and other details on our numerical experiments have been provided in Suppl. Section A.

## Acknowledgments

NHK acknowledges the support of scholarships from UNIQUE (`https://www.unique.quebec`) and IVADO (`https://ivado.ca`). DL acknowledges support from the FRQNT Strategic Clusters Program (2020-RS4-265502 – Centre UNIQUE – Unifying Neuroscience and Artificial Intelligence – Québec) and the Richard and Edith Strauss Postdoctoral Fellowship in Medicine. BAR acknowledges support from NSERC (Discovery Grant: RGPIN-2020-05105; Discovery Accelerator Supplement: RGPAS-2020-00031; Arthur B McDonald Fellowship: 566355-2022) and CIFAR (Canada AI Chair; Learning in Machine and Brains Fellowship). GL acknowledges support from NSERC (Discovery Grant: RGPIN-2018-04821), CIFAR (Canada AI Chair), and the Canada Research Chair in Neural Computations and Interfacing. The authors also acknowledge the support of computational resources provided by Mila (`https://mila.quebec`) and NVIDIA that enabled this research.

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

# SUPPLEMENTARY MATERIAL

## A  NUMERICAL SIMULATION DETAILS

### A.1  TASK HYPERPARAMETERS

Table A.1: Hyperparameters for the spatial position estimation task.

| Hyperparameter | Value |
|---|---|
| TASK | |
| Environment size | $2.2\,\text{m} \times 2.2\,\text{m}$ |
| Border region | $0.03\,\text{m}$ |
| Border slowdown factor | $0.25$ |
| Position initialization | $\mathcal{U}(-2.2, 2.2)$ |
| Rotation velocity bias | $0\,\text{rad/s}$ |
| Rotation velocity std. dev. | $11.52\,\text{rad/s}$ |
| Rayleigh forward velocity | $0.2\,\text{m/s}$ |
| Biasing anchor point | $(0, 0)$ |
| Biasing drift constant | $0.05$ |
| # place cells | $512$ |
| $\sigma_{SP}$ | $0.2$ |
| Sequence length | active $= 100$, quiescent $= 200$ |
| $\sigma$ | $\frac{0.01}{\sqrt{0.02}} \approx 0.0707$ |
| NETWORK | |
| # recurrent units | $512$ |
| $\tau$ | $0.1$ |
| TRAINING | |
| Batch size | $200$ |
| # batches | $2500$ |
| Optimizer | Adam |
| Learning rate | $0.001$ |

Table A.2: Hyperparameters for the head direction estimation task.

| Hyperparameter | Value |
|---|---|
| TASK | |
| Bearing initialization | $\mathcal{U}(-\pi, \pi)$ |
| Rotation velocity bias | $0\,\text{rad/s}$ |
| Rotation velocity std. dev. | $11.52\,\text{rad/s}$ |
| # head direction cells | $512$ |
| $\sigma_{HD}$ | $\frac{\pi}{6}$ |
| Sequence length | active $= 100$, quiescent $= 200$ |
| $\sigma$ | $\frac{0.1}{\sqrt{0.02}} \approx 0.7071$ |
| NETWORK | |
| # recurrent units | $128$ |
| $\tau$ | $0.04$ |
| TRAINING | |
| Batch size | $200$ |
| # batches | $20000$ |
| Optimizer | Adam |
| Learning rate | $0.001$ |

## A.2 ADDITIONAL DETAILS

**Kernel Density Estimation (KDE).**   To compute KDEs, we used the `stats.gaussian_kde()` method from `scipy` (Virtanen et al., 2020), with all hyperparameters set to their default values. For all KDE plots, we computed the density estimates using 200 trajectories of 100 timesteps each for the active phase and 200 timesteps each for the quiescent phase. For Fig. 2e, we estimated densities using 200 trajectories of 100 timesteps each for the active phase and 1000 timesteps each for the quiescent phase (to get a better estimate of the steady-state distribution). We used 2500 samples from each distribution to compute the KL divergence by Monte Carlo approximation.

**Quantifying exploration during quiescence.**   To analyze the importance of noise for exploration during quiescence, we computed the steady-state total variance (variance summed over output dimensions) for 200 quiescent trajectories from each trained network. For Fig. 2f, we generated trajectories of 2000 timesteps each and truncated the first 1000 timesteps before computing the variance, to demonstrate that noise facilitates greater and continued exploration over long timescales.

## A.3 NOISY CONTINUOUS-TIME RNNS

**"Vanilla" RNNs.**   For most of our numerical experiments, we use noisy "vanilla" continuous-time RNNs with linear readouts. The equations for network updates and output estimates are as follows:

$$\Delta\mathbf{r}(t) = \frac{1}{\tau}\big[-\mathbf{r}(t) + \text{ReLU}\big(\mathbf{W}^{rec}\mathbf{r}(t) + \mathbf{W}^{in}\Delta\mathbf{s}(t)\big)\big]\Delta t + \sigma\boldsymbol{\eta}(t) \tag{A.1}$$

$$\mathbf{o} = \mathbf{D}\mathbf{r}(t), \tag{A.2}$$

where $\mathbf{r}(t)$ represents the network activity at time $t$, $\Delta\mathbf{s}(t)$ is a change-based input to the network, $\mathbf{W}^{rec}$ and $\mathbf{W}^{in}$ are the recurrent and input weight matrices respectively, $\tau$ is the RNN time constant, and $\boldsymbol{\eta}(t) \sim \mathcal{N}(0, \Delta t)$ is Brownian noise. The continuous-time dynamics are approximated using the Euler-Maruyama method with integration timestep $\Delta t = 0.02\,\text{s}$. The network's activity is transformed by a linear mapping $\mathbf{D}$ to predicted place cell or head direction cell activities $\mathbf{o}$. During the quiescent phase, we simulated network activity in the absence of stimuli ($\Delta\mathbf{s}(t) = 0$), and doubled the noise variance, as prescribed by our mathematical analysis (Section 2.4).

We set the value of $\tau$ for each task to ensure that the RNN is able to respond quick enough to inputs and ensure optimal performance. Further, for each task we choose different training values for $\sigma$ to scale the Brownian noise to establish an effective signal-to-noise ratio that is high enough to accurately solve the task. From this baseline noise level, quiescent trajectories were calculated with doubled variance.

**Gated recurrent units (GRUs).**   We also used a continuous-time GRU formulation similar to the one described by Jordan et al. (2021). The equations for network updates and output estimates are:

$$\mathbf{z}(t) = \text{sigmoid}\big(\mathbf{W}_z^{rec}\mathbf{r}(t) + \mathbf{W}_z^{in}\Delta\mathbf{s}(t)\big) \tag{A.3}$$

$$\mathbf{g}(t) = \text{sigmoid}\big(\mathbf{W}_g^{rec}\mathbf{r}(t) + \mathbf{W}_g^{in}\Delta\mathbf{s}(t)\big) \tag{A.4}$$

$$\Delta\mathbf{r}(t) = \frac{1}{\tau}\big[\big(1 - \mathbf{z}(t)\big) \odot \big(\tanh\big(\mathbf{W}_r^{rec}\big(\mathbf{g}(t) \odot \mathbf{r}(t)\big) + \mathbf{W}_r^{in}\Delta\mathbf{s}(t)\big) - \mathbf{r}(t)\big)\big]\Delta t + \sigma\boldsymbol{\eta}(t) \tag{A.5}$$

$$\mathbf{o} = \mathbf{D}\mathbf{r}(t), \tag{A.6}$$

where $\mathbf{r}(t)$ represents the network activity at time $t$, $\mathbf{z}(t)$ and $\mathbf{g}(t)$ are gates, $\Delta\mathbf{s}(t)$ is a change-based input to the network, $\mathbf{W}_z^{rec}$, $\mathbf{W}_z^{in}$, $\mathbf{W}_g^{rec}$, $\mathbf{W}_g^{in}$, $\mathbf{W}_r^{rec}$ and $\mathbf{W}_r^{in}$ are weight matrices, $\tau$ is the time constant, and $\boldsymbol{\eta}(t) \sim \mathcal{N}(0, \Delta t)$ is Brownian noise. The continuous-time dynamics are approximated using the Euler-Maruyama method with integration timestep $\Delta t = 0.02\,\text{s}$. Just as with the vanilla RNN, the network's activity is transformed by a linear mapping $\mathbf{D}$ to predicted place cell or head direction cell activities $\mathbf{o}$. During the quiescent phase, we simulated network activity in the absence of stimuli ($\Delta\mathbf{s}(t) = 0$), but we do not double the noise variance due to noise-sensitivity. We set appropriate values for $\tau$ and $\sigma$, just as with the vanilla RNNs.

# B    KEY ASSUMPTIONS FOR MATHEMATICAL RESULTS

1. We consider discrete-time approximations of noisy continuous-time RNNs. Using continuous-time RNNs is a common practice in the literature, and we use their discrete-time approximations to be in line with our numerical simulations.

2. The network must be performing some variant of path integration, by integrating change-based information about environmental state variables to some function of these variables. This condition is often met in natural settings, such as spatial navigation.

3. We assume that the inputs to the network are drawn from a stationary distribution, which amounts to ignoring the effects of initial conditions on state occupancy statistics, and also assuming that the behavioral policy remains constant throughout time.

4. We consider greedy optimization of the loss at every timestep. Greedy optimization is a sensible way of partitioning effort across time in this task: the network does the best that it can at each timestep, assuming that at each previous timestep the best possible job has been done. In the absence of noise, the greedily optimal solution is equivalent to path integration, which is also a *globally* optimal solution.

5. We assume that the network is performing optimally in the presence of noise. In practice, we train our networks until their loss reaches very low, near-zero values.

6. We assume that our network dynamics can be decomposed into two terms with different functional dependencies. The first term depends only on the activity, while the second term depends on all original dependencies—the activity, the state, and the change-based inputs.

7. For the quiescent phase, we assume that the change-based sensory inputs are zero. This is reasonable because for tasks like those we have considered, which involve integrating self-motion cues, $\frac{d\mathbf{s}(t)}{dt}$ must be zero during periods of quiescence like sleep, where the animal is not moving and hence does not receive sensory inputs associated with self-motion.

8. While we assume that the noise variance is doubled during quiescence to show *exact* equivalence with Langevin sampling, this is by no means *necessary* to witness reactivation (Suppl. Fig. C.2). This noise variance is equivalent to a temperature parameter for the sampling.

# C   ADDITIONAL NUMERICAL SIMULATIONS

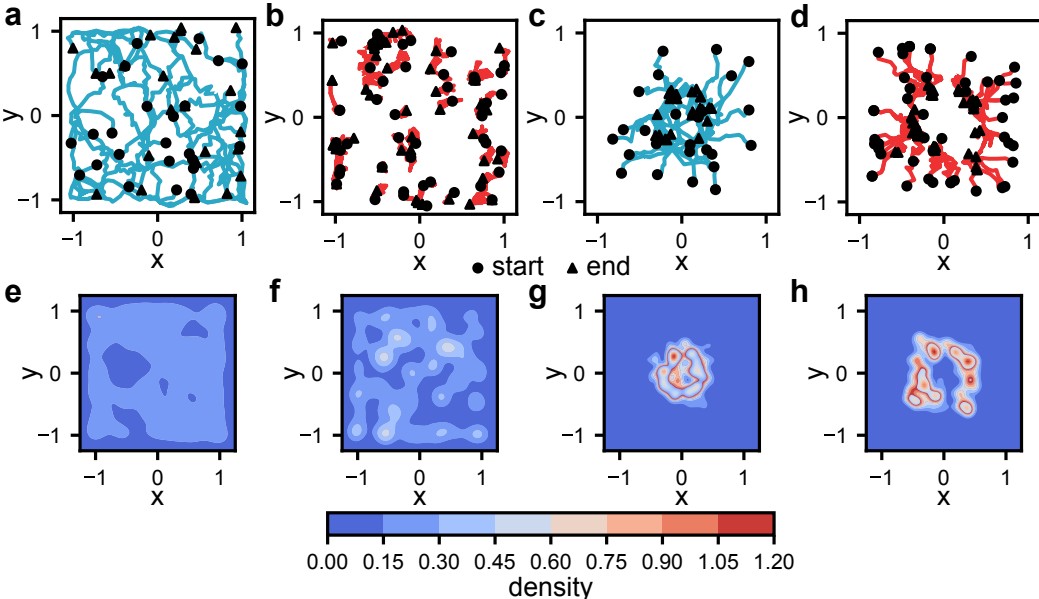

Figure C.1: **Spatial position estimation results without increased noise variance during quiescence. a-b)** Decoded output trajectories during active (a) and quiescent (b) phases for a network trained on the unbiased task. **c-d)** Same as (a-b) but for a network trained on the biased task. **e-f)** KDE plots for 200 decoded active (e) and quiescent (f) output trajectories for the unbiased task. **g-h)** Same as (e-f) but for the biased task.

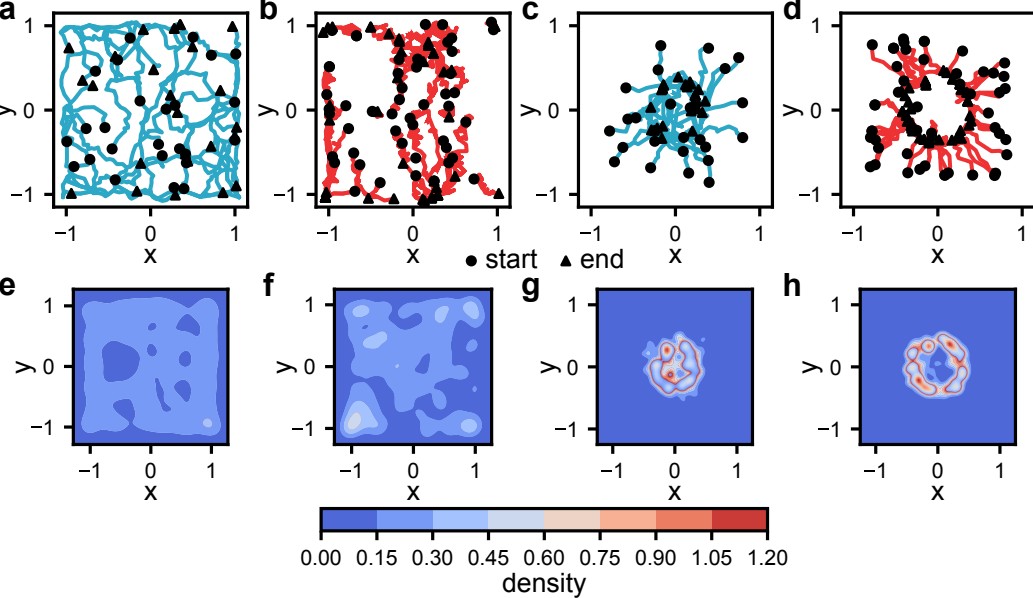

Figure C.2: **Spatial position estimation results for GRU networks. a-b)** Decoded output trajectories during active (a) and quiescent (b) phases for a network trained on the unbiased task. **c-d)** Same as (a-b) but for a network trained on the biased task. **e-f)** KDE plots for 200 decoded active (e) and quiescent (f) output trajectories for the unbiased task. **g-h)** Same as (e-f) but for the biased task.

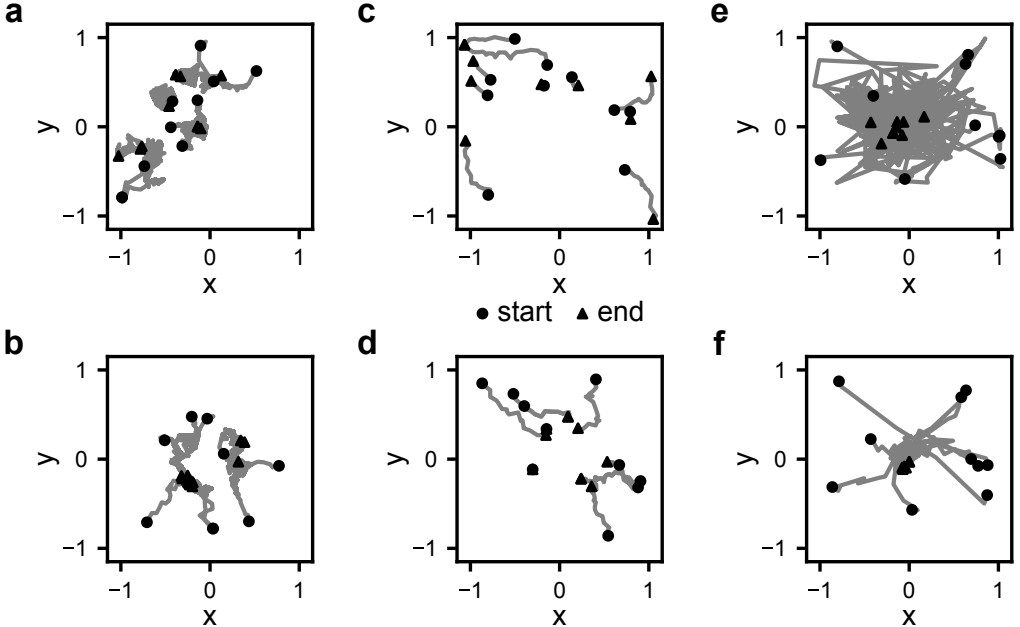

Figure C.3: **Example decoded quiescent trajectories under different noise conditions for spatial position estimation. a-b)** Example noisy quiescent trajectories for a network trained in the presence of noise, for the unbiased (a) and biased (b) tasks. **c-d)** Same as (a-b), but for noiseless quiescent trajectories for a network trained without noise. **e-f)** Same as (a-b), but for noisy quiescent trajectories for a network trained without noise.

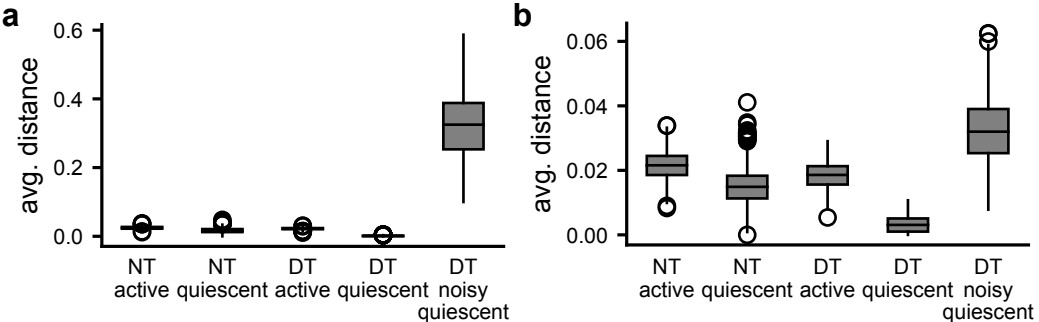

Figure C.4: **Distributions of average distances between consecutive points in output trajectories for unbiased and biased spatial position estimation. a)** Distributions for the unbiased task. NT denotes noisy training while DT denotes deterministic (noiseless) training. **b)** Same as (a) but for the biased task. In each case, the box and whisker plots show the distribution of within-trajectory average point-to-point distances, computed for 200 output trajectories and 5 random seeds.

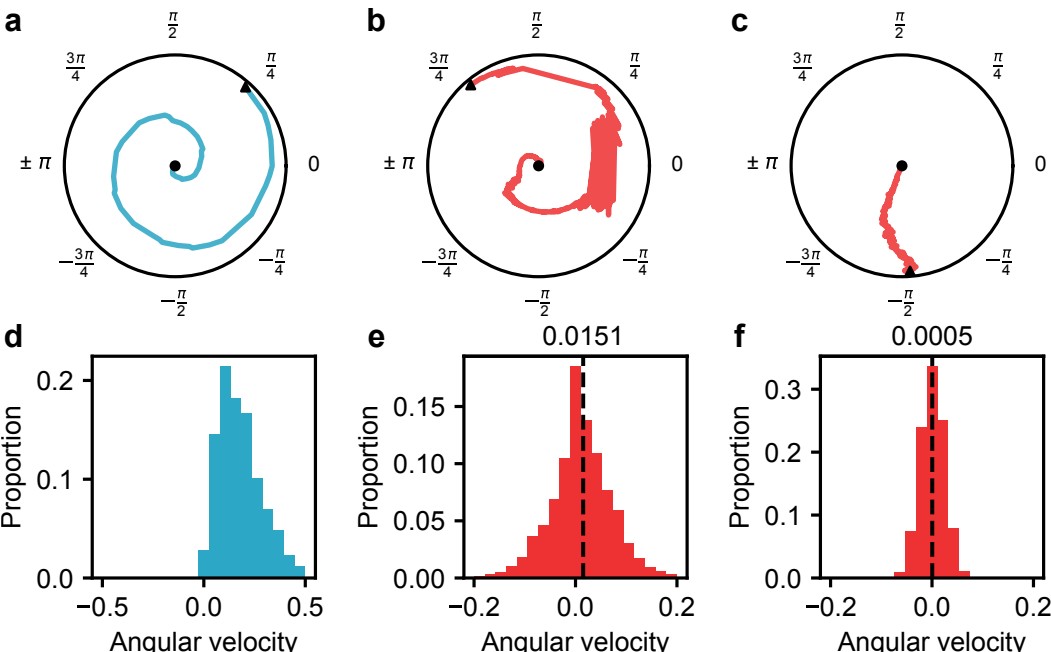

Figure C.5: **Biased moment-to-moment transition structure for the head direction estimation task. a-b)** Decoded trajectories from the active (a) and quiescent (b) phases, for a biased network trained on counter-clockwise trajectories. **c)** Decoded trajectory from the quiescent phase for an unbiased network. Sequence length is 50 timesteps for active phase trajectories (a) and 500 timesteps for quiescent trajectories (b-c). **d-f)** Distributions of angular velocities during the active (d) and quiescent (e) phases for a biased network, and the quiescent phase for an unbiased network (f). Dashed line denotes the mean.

