# OpenReview forum: "Sufficient conditions for offline reactivation in recurrent neural networks"
_ICLR.cc/2024/Conference — ICLR 2024 poster_

### Official Review · Reviewer_NzXs · 2023-10-29

**Soundness:** 3 good
**Presentation:** 2 fair
**Contribution:** 2 fair
**Rating:** 6
**Confidence:** 5

**Summary:**

The paper proposes a training algorithm for continuous-time RNN integration in the presence of noise that relies on an upper bound for the loss that is split into a denoising and a tracking part. It is then shown for a angular and a motion velocity integration task that the noise compensation dynamics induce diffusive reactivations in a quiescent state.

**Strengths:**

The paper introduces a novel method of task-optimization for RNNs that rely on an upper bound for the loss that is split into two parts: the first term optimizes for denoising and the second term is optimized for tracking the signal. This is the first paper that discusses the contribution of learning in the presence of noise to have implication to reactivations.
Through this framework a new perspective is proposed on reactivations in the brain being the consequence of the presence of noise.
Finally, this framework provides interpretability to the system that solves an integration task in terms of what part of the dynamics corresponds to denoising the systems and which to integrating instantaneous changes of the state variable.
The effectiveness of the approach is demonstrated against a couple of integration tasks highlighting its applicability. The theorems and proofs are sound.
Furthermore, the paper is well written and the contributions are clearly explained with possible implications of the work for neuroscience.
These implications for neuroscience might provide a new perspective on how neural dynamics that implements neural integration might be decomposed into a denoising and an integrating component and how networks that learn in the presence of noise then exhibit reactivations.

**Weaknesses:**

It is unclear what is meant with tuning of the value of $\tau$. Is this for a fixed noise level? Or is it chosen as the optimal for all used noise levels for training?

\paragraph{Figures}
Use some transparency in Figure 1 f and e so that overlapping points are visible. At the moment it is unclear what proportion of the points is hidden behind the first layer and what their distribution is.


\paragraph{Contribution}
The the demonstration contribution of this framework could benefit from some additional experiments.
Comparison to training without noise and added noise in quiescent state is missing.
Comparison to other ring attractor/head direction models is missing.

Finally, some of the implications of the proposed framework require more justification.
The conclusion that reverse replay could be explained by diffusive reactivations seems like a stretch and should be substantiated better.
Also, reactivations such as replay are not the type of reactivations that are found in the networks in the paper (this is admitted in the paper) even though the introduction is discussing those for a large part.

\paragraph{Noise}
The fact that training without noise resulted in erratic output trajectories might be explained by the statistics of the input.
Does the used input static reflect best what happens in animal behavior?
It would be good to compare the statistics of state sequences and reactivation sequence on the level of the sequences themselves.
Because in terms of optimality of exploration erratic trajectories might be more optimal to fully explore, see for example McNamee (2021). But see also Supplementary Figure A.1 (a) vs (e) that seems to show that a network that  has been trained without noise (e) explores a bigger part of the state space than a network that has been trained in the presence of noise (a).

Finally, the claim that even with the addition of noise the failure of exploration could not be corrected (page 8, middle) should be better substantiated with a comparison based on distributions rather than just example trajectories (currently Suppl.Fi. A.1e-f is used as justification).


McNamee, D.C., Stachenfeld, K.L., Botvinick, M.M. and Gershman, S.J., 2021. Flexible modulation of sequence generation in the entorhinal–hippocampal system. Nature neuroscience, 24(6), pp.851-862.

**Questions:**

How is the bias exactly defined for figure 2?
In Figure 2, what distribution are random networks sampled from? Further, what does it mean that a random network s trained and tested in the presence of noise (as would be implied in the column $\text{R}^\sigma$).


Why is the kernel density estimate the best (or a good) way to compare trained networks?

How does the final trained network in terms of its parameters compare to the full greedily optimal dynamics?
How do different trained networks compare to each other in terms of the reactivation statistics?

Is it truly a ring attractor that solves the angular integration task? On the given time scale it could be the case that the solution is provided by a line attractor that gets mapped to the ring by $D$.

In Equation (17) is the optimum unique?

---

> ### Author Response · Authors · 2023-11-17
> **Response to Reviewer NzXs**
>
> We would like to thank the reviewer for their review and helpful suggestions. We appreciate that the reviewer found our theory to be novel, providing interpretability, and an interesting new perspective on reactivations. We are glad that the reviewer found the paper to be well-written and our contributions clearly explained. We have addressed the points and questions in the review below:
>
> > It is unclear what is meant with tuning of the value of \tau. Is this for a fixed noise level? Or is it chosen as the optimal for all used noise levels for training?
>
> **Response:** We choose $\tau$ for the RNN based on the task to ensure it performs optimally, and it remains the same for all noise levels. The intention is to ensure that the network is faster than the inputs or else it does not react quick enough to new inputs, causing the activity to remain stuck and the performance to be poor. In practice, we do not have to tune $\tau$ very carefully, we set it to a reasonable value that ensures that the network's dynamics are fast enough to process the task's inputs and perform optimally. We clarify this point in the revised manuscript.
>
> > Use some transparency in Figure 1 f and e so that overlapping points are visible. At the moment it is unclear what proportion of the points is hidden behind the first layer and what their distribution is.
>
> **Response:** We apologize if the plot was not clear. We tried adding transparency to these plots but they did not aid in visualizing the distribution better, due to the interaction with different shades of color that attempt to show the structure of the attractor manifold, and its correspondence with the environment.
>
> Nevertheless, this is why we provide the kernel density estimates below! The KDE provides an actual distribution on the data, i.e., a principled visualization of the distributions, rather than requiring visual estimation based on occupied points. We hope that this plot aids the reviewer in comparing the distributions.
>
> > The the demonstration contribution of this framework could benefit from some additional experiments. Comparison to training without noise and added noise in quiescent state is missing. Comparison to other ring attractor/head direction models is missing.
>
> **Response:**  We apologize if this was unclear from the text. We have compared to training without noise and with added noise in quiescent state in Suppl. Fig. A.2e-f. We have shown in this experiment that these trajectories are erratic, and do not properly correct for lack of exploration arising from training without noise. In an additional experiment, we have shown that the point-to-point distances are much higher for these erratic trajectories than even waking behavior, indicating their implausibility.
>
> It is true that we don't compare to more traditional ring attractor models, with which we have some key differences. The critical distinction is that our model recapitulates statistics of waking navigation, whereas networks with hand-tuned weights, for instance, would only be able to produce a uniform distribution over the attractor manifold in question.
>
> > Finally, some of the implications of the proposed framework require more justification. The conclusion that reverse replay could be explained by diffusive reactivations seems like a stretch and should be substantiated better. Also, reactivations such as replay are not the type of reactivations that are found in the networks in the paper (this is admitted in the paper) even though the introduction is discussing those for a large part.
>
> **Response:** We apologize if our claims were not clear. In our mathematical results, we are only claiming that the steady-state distribution of quiescent activity matches that of waking activity. Specifically, we do not make any comment on the sequential dynamics of quiescent trajectories themselves. Thus, for reverse replay, the only point we make is that for true diffusive rehearsal, reversed trajectories are as probable as forward trajectories. In additional experiments with our RNNs on the head direction task (Suppl. Fig. A.5), we biased the waking moment-to-moment transition structure to be counter-clockwise. We found that quiescent trajectories partially recapitulated the transition structure of active phase trajectories, and maintained their bearing for longer periods, resembling real neural activity more closely. The biased velocities were also reflected during quiescence, although less clearly than waking. We do not yet have theoretical justification for this. However, we note that reversals in the trajectories still remain possible and do occur (Suppl. Fig. A.5).

---

> > ### Author Response · Authors · 2023-11-17
> > **Response to Reviewer NzXs**
> >
> > > The fact that training without noise resulted in erratic output trajectories might be explained by the statistics of the input. Does the used input static reflect best what happens in animal behavior? It would be good to compare the statistics of state sequences and reactivation sequence on the level of the sequences themselves. Because in terms of optimality of exploration erratic trajectories might be more optimal to fully explore, see for example McNamee (2021). But see also Supplementary Figure A.1 (a) vs (e) that seems to show that a network that has been trained without noise (e) explores a bigger part of the state space than a network that has been trained in the presence of noise (a).
> >
> > **Response:** We thank the reviewer for this question. Our data generation protocol is identical to influential prior work (Sorscher et al., 2019; Banino et al., 2018) which generate rat motion trajectories based on the model described in yet another influential work (Erdem & Hasselmo, 2012). As our work is consistent with these prior works and uses the same data generation as Sorscher et al., 2019, we believe it reflects natural animal behavior.
> >
> > We would like to point out that the trajectories arising from noisy reactivations after noiseless training are not 'exploring' more of the state space, they are truly discontinuous and erratic, and violate the actual map that was formed. Thus, it is hard to see how these could be useful. This is evident from Suppl. Fig. A.3, where we see that the distances between consecutive points in the noisy quiescent trajectories with noiseless training is much higher than those for all other conditions including active behavior.
> >
> > We also carried out an experiment with the head direction task to comment more on the trajectory-level statistics. We trained our networks solely on counter-clockwise trajectories for the head direction task (Suppl. Fig. A.5) and found that the reactivations recapitulated the transition structure of active phase trajectories. They also maintained their bearing for longer periods, resembling real neural activity more closely. The biased velocities were reflected in quiescent trajectories, but less clearly than during waking. This shows that trajectory level features of active behavior are reflected during quiescence.
> >
> > Sorscher et al. A unified theory for the origin of grid cells through the lens of pattern formation. Neural Information Processing Systems (2019).
> >
> > Banino et al. Vector-based navigation using grid-like representations in artificial agents. Nature 557 (2018): 429 - 433.
> >
> > Erdem & Hasselmo. A goal-directed spatial navigation model using forward trajectory planning based on grid cells. European Journal of Neuroscience, 35: 916-931. 2012.
> >
> > > Finally, the claim that even with the addition of noise the failure of exploration could not be corrected (page 8, middle) should be better substantiated with a comparison based on distributions rather than just example trajectories (currently Suppl.Fi. A.1e-f is used as justification).
> >
> > **Response:** We thank the reviewer for this useful suggestion. We have added Suppl. Fig. A.3 which shows exactly this by comparing the distribution of distances between successive points in a trajectory across conditions. We find that this distance is much higher for quiescent trajectories with added noise without noise in training, than even waking behavior. This indicates simply the implausibility of these trajectories.

---

> > > ### Author Response · Authors · 2023-11-17
> > > **Response to Reviewer NzXs**
> > >
> > > > How is the bias exactly defined for figure 2? In Figure 2, what distribution are random networks sampled from? Further, what does it mean that a random network s trained and tested in the presence of noise (as would be implied in the column R^\sigma).
> > >
> > > **Response:**
> > >
> > > 1. The bias is defined by adding a drift term towards an anchor point, such as the center of the environment. When we generate a random trajectory to train on, we generate a series of random turns and movements for each timestep. The bias/drift is defined as the displacement vector between the current location and the anchor point, scaled down by a constant. Thus, for each timestep, we have:
> > >     ```
> > >     new_position = old_position + drift_constant * (anchor_point - old_position) + random_movement
> > >     ```
> > >     This ensures that the positions occupied are values close to the anchor point.
> > >
> > > 2. The random untrained networks' weights are initialized using the PyTorch default, i.e., each of the recurrent and readout weights is sampled from $\mathcal{U}(-\frac{1}{\sqrt{N_r}}, \frac{1}{\sqrt{N_r}})$, where $N_r$ is the dimensionality of network activity, and each of the input weights is sampled from $\mathcal{U}(-\frac{1}{\sqrt{N_s}}, \frac{1}{\sqrt{N_s}})$, where $N_s$ is the dimensionality of input stimuli.
> > > 3. Random networks are untrained, and testing in the presence of noise means that noise is added to the activity during the quiescent phase. We thank the reviewer for pointing us to the incorrect framing of the caption. We have modified the caption to avoid misunderstandings in the updated draft.
> > >
> > > > Why is the kernel density estimate the best (or a good) way to compare trained networks?
> > >
> > > **Response:** We believe that kernel density estimation (KDE) is a good tool for our analyses. KDE is a non-parametric density estimation method, so it makes almost no assumptions on the specifics of the underlying distribution, unlike parametric methods. This is important as we do not want to make assumptions on the quiescent phase output distribution. The KDE gives us a probability density function and allows for easy sampling, which allows us to compute KL-divergences (by Monte Carlo simulations) easily for our analyses. Finally, KDE provides for intuitive visualizations of the distributions to drive across the main idea of our analyses. The only issue is that KDE is data-hungry, but this is fine for low dimensions as in our case.
> > >
> > > > How does the final trained network in terms of its parameters compare to the full greedily optimal dynamics? How do different trained networks compare to each other in terms of the reactivation statistics?
> > >
> > > **Response:** Evaluating how the network compares to the greedily optimal dynamics is difficult because we do not have access to $p(\mathbf{r})$. Thus, it would require us to use multiple approximations to compute $\nabla\log p(\mathbf{r})$ for the greedily optimal dynamics, and so it would introduce errors in our estimate of the ground-truth optimal behavior. We did validate that our networks behave in the quiescent phase as Langevin sampling should by comparing the steady-state statistics of quiescent behavior with active behavior, and we see that they do indeed match.
> > >
> > > Different networks would have different individual neurons, so it is not possible to directly compare the activity of different trained networks. However, different networks trained on the same task statistics (i.e., unbiased or biased) have similar attractor manifolds, and show similar behavioral outputs during quiescence. For example, two different RNNs trained on the biased navigation task will have similar steady-state distributions of output trajectories, since the task statistics are identical, with the only differences arising from samples of randomly generated training data (the individual samples may differ but the distribution is identical) and the values of noise sampled.
> > >
> > > > Is it truly a ring attractor that solves the angular integration task? On the given time scale it could be the case that the solution is provided by a line attractor that gets mapped to the ring by D.
> > >
> > > **Response:** We indeed believe that a ring attractor is required to solve the task. Our network learned a ring attractor without issue, and without having to enforce this attractor structure in its dynamics. While a line attractor could provide a local approximation of the ring attractor, it would be unable to match the nonlinear topology of the full ring---the output has periodic boundary conditions, and a linear decoder could not warp a line attractor such that it has periodic boundary conditions. We initialized randomly our initial head bearing, so the local approximation would have to be accurate across the whole manifold, irrespective of the time scale.

---

> > > > ### Author Response · Authors · 2023-11-17
> > > > **Response to Reviewer NzXs**
> > > >
> > > > > In Equation (17) is the optimum unique?
> > > >
> > > > **Response:** This is the minimum norm solution, but in general, $\mathbf{D}$ is mapping a higher-dimensional activity space to a lower-dimensional output space, which means that there are extra degrees of freedom in the RNN dynamics that are not constrained by the task. So this optimum is not unique.
> > > >
> > > > We believe we have addressed all of the reviewer's concerns, and would be grateful if the reviewer would consider raising their score. We would be happy to discuss further if the reviewer has any additional concerns. We eagerly await the reviewer's response.

---

> ### Author Response · Authors · 2023-11-20
> **Following up with the reviewer**
>
> Dear Reviewer NzXs,
>
> We believe our response has addressed all of the reviewers' concerns and would be grateful if the reviewer would consider raising their score. As the discussion period is coming to an end soon, we would greatly appreciate a response from the reviewer. If the reviewer has any further questions, we would be happy to address them. We thank the reviewer again for their helpful suggestions and eagerly look forward to a response.

---

> ### Author Response · Authors · 2023-11-22
> **Discussion period coming to an end**
>
> Dear Reviewer NzXs,
>
> This is a gentle reminder that the discussion period with authors is coming to an end (<20 hours). We believe our responses in the comments and updated manuscript have addressed all of the reviewers' concerns, and we would be grateful if the reviewer would consider raising their score. If the reviewer has any further questions, we would be happy to address them in the short time remaining. We thank the reviewer again for their helpful suggestions and eagerly look forward to a response.

---

> > ### Comment · Reviewer_NzXs · 2023-11-22
> >
> > Thank you for the response. I appreciate the additional details and the new version is taking steps in the right direction.  I have raised my score.
> >
> > Some remaining concerns:
> > >  for reverse replay, the only point we make is that for true diffusive rehearsal, reversed trajectories are as probable as forward trajectories.
> >
> > But both would be very unlikely compared to diffusion type dynamics.
> >
> >
> >
> > > Is it truly a ring attractor that solves the angular integration task?
> >
> > The authors claim that "the networks learned a ring attractor without issue" but it is unclear how that is
> >
> > How do you know that the network has periodic boundary conditions? It seems that the networks has only been run on finite time trials, which could leave open the opportunity of a (long) curved line attractor being mapped onto a ring.

---

> > > ### Author Response · Authors · 2023-11-23
> > > **Response to Reviewer NzXs**
> > >
> > > We greatly appreciate the reviewer's response to our comments. We are glad that the reviewer finds our new analyses to be steps in the right direction. We thank the reviewer for their comments and suggestions that have been instrumental in improving our work, and also for the increase in score. Here, we address the reviewer's remaining concerns:
> > >
> > > > But both would be very unlikely compared to diffusion type dynamics.
> > >
> > > We would like to clarify that we do not impose any constraints on the dynamics of the trajectories. Thus, in our case, it is evident that forward and reversed trajectories could both occur during quiescence. This is clearly visible in Suppl. Fig. A.5b, where during quiescence, the moment-to-moment transition structure is partially recapitulated, but there are still portions of the trajectory that are reversed, i.e., do not share the same transition structure. Furthermore, diffusion-like trajectories are not unreasonable to expect from a physiological point of view---there is experimental evidence of diffusive dynamics in the hippocampus (Stella et al., 2019) and head direction system (Chaudhuri et al., 2019---Fig. 4).
> > >
> > > Stella et al. Hippocampal Reactivation of Random Trajectories Resembling Brownian Diffusion. Neuron vol. 102,2: 450-461.e7 (2019).
> > >
> > > Chaudhuri et al. The intrinsic attractor manifold and population dynamics of a canonical cognitive circuit across waking and sleep. Nat Neurosci 22, 1512–1520 (2019).
> > >
> > > > The authors claim that "the networks learned a ring attractor without issue" but it is unclear how that is
> > > >
> > > > How do you know that the network has periodic boundary conditions? It seems that the networks has only been run on finite time trials, which could leave open the opportunity of a (long) curved line attractor being mapped onto a ring.
> > >
> > > Firstly, we would like to clarify that it is not a major claim of this paper that the network has learnt a ring attractor. However, we would like to point out to Fig. 3c-d, which clearly show the ring attractor structure of the RNN's activity. We did not impose any constraints on the network that forced it to learn a ring attractor. The network in this case has clearly learnt a periodic solution which is consistent with the task structure (head direction is between $-\pi$ and $\pi$). Several previous works have also shown ring attractors to be learnt by RNNs trained on this task (Cueva et al. 2020---Fig. 10; Vafidis et al., 2022).
> > >
> > > Cueva et al. Emergence of functional and structural properties of the head direction system by optimization of recurrent neural networks. ICLR (2020).
> > >
> > > Vafidis et al. Learning accurate path integration in ring attractor models of the head direction system. eLife 11:e69841 (2022).
> > >
> > > We believe that we have addressed the reviewer's remaining concerns, and would be grateful if the reviewer could consider an increase in their rating in light of our response. We thank the reviewer once again and eagerly await their response.

---

### Official Review · Reviewer_aZpo · 2023-10-30

**Soundness:** 2 fair
**Presentation:** 3 good
**Contribution:** 3 good
**Rating:** 6
**Confidence:** 4

**Summary:**

The authors demonstrate that training network dynamics on an upper bound of the cost function for learning a function of an input stimulus in the presence of network noise results in network dynamics that separate into a denoising component and a signal integration component.  Furthermore, they provide conditions such that, in the absence	of input, the network will reproduce typical states of the (learned function of the) input stimulus.  Notably, this strictly requires the input noise to be different in the quiescent vs input-driven cases.  The authors demonstrate numerical simulations in which networks reflect these results.

**Strengths:**

This paper does a nice job of outlining how the cost function can be deconstructed into two components, the denoising component and the signal integration component.  The idea of replay being a result of signal integration in the presence of noise is an interesting one, and the authors present a simple mechanism whereby this can arise. This is a contribution to an important question in neuroscience.  The theoretical approach, results, and the numerical experiments are well described and supported.

**Weaknesses:**

The paper is potentially confusing to some readers in that the presentation (and the numerical experiments if I have followed) regard training a network with a fixed nonlinearity and trainable weight parameters (i.e. a typical RNN) but the theoretical analysis considers not a fixed nonlinearity, but rather the space of all possible dynamics that could govern the system.  I suspect this may misdirect some readers regarding the basic argument (possibly me as well).

Training a regular RNN on the cost function, L, of course, does not guarantee anything about the cost function L_upper, so one must make some additional assumptions about the structure of the network dynamics, namely something akin to demanding that the space of permissible dynamics admits the derived minimum of L_upper.  In the case of the numerical experiments, the authors use ReLU and decompose the input to the nonlinearity into a recurrent part and a signal integration part, which should approximately allow for the decomposition that leads to L_upper.  One needs to add an argument that this decomposition is reasonable for certain classes of RNNs. E.g. If the network is in the active regime, the ReLU doesn't contribute much and then the composition amounts to considering the activity updates from the recurrent part and the input part separately.  One could make a similar argument about sigmoids.  In short, the authors should, I think, be direct about the assumptions on the RNN necessary for their theory.

Regarding the quiescent state, I think the authors are perhaps a little too cavalier with the fact that, strictly speaking, the noise term must change in order to guarantee replay.  As presented at the moment, it feels to this reader like a weakness that is not adequately addressed.  The authors should either supply 1) a reason to believe this sort of change happens in real networks or 2) a demonstration that the deviations are not sufficient to qualitatively change the main result.  There is also the possibility that this is an opportunity for the authors.  The deviations from perfect replay if the noise does not change amount to a prediction, and one which may have computational advantages (e.g. facilitating exploration in terms of a planning algorithm).

**Questions:**

Last sentence of the introduction:  I think "ecologically" should be "ethnologically"

Questions:

- Does this work for more general RNNs (more general nonlinearities, interactions between recurrence and signal integration, etc.)?

- Is there reason to believe the noise distribution changes in the quiescent state?

---

> ### Author Response · Authors · 2023-11-17
> **Response to Reviewer aZpo**
>
> We would like to thank the reviewer for their review and helpful suggestions. We appreciate that the reviewer found our theory and experiments to be interesting, well-described and well-supported. We have addressed the points and questions in the review below:
>
> > The paper is potentially confusing to some readers in that the presentation (and the numerical experiments if I have followed) regard training a network with a fixed nonlinearity and trainable weight parameters (i.e. a typical RNN) but the theoretical analysis considers not a fixed nonlinearity, but rather the space of all possible dynamics that could govern the system. I suspect this may misdirect some readers regarding the basic argument (possibly me as well).
>
> **Response:** We apologize if this was not clear. The idea is that the RNNs trained optimally are approximating the ground-truth optimal solution, given by the greedily optimal dynamics (Section 2.3). It is indeed reasonable for an RNN to implement Eq. 18—according to the literature on RNNs being universal function approximators (Funahashi & Nakamura, 1993). We are thus assuming that our trained RNNs are near the optimum, and our empirical results indicate that they are. Our theoretical results do not make any assumptions about the architecture of the RNN (in fact we have added simulations with GRUs, Suppl. Fig. A.4), only that it is trained to be optimal. Indeed we see the Langevin sampling during quiescence leading to the distributional match. It is certainly possible for suboptimal (or very small) RNNs to fail to align with the theory, the theory only applies for very well-trained networks that are performing near-optimally such as our networks.
>
> Funahashi & Nakamura. Approximation of dynamical systems by continuous time recurrent neural networks. Neural Networks 6 (1993): 801-806.
>
> > Training a regular RNN on the cost function [...] In short, the authors should, I think, be direct about the assumptions on the RNN necessary for their theory.
>
> **Response:** We apologize if this was not clear. We would like to state that our mathematical analysis is a theory for high capacity networks capable of legitimately approximating the optimal solution. RNNs are capable of doing this, as they are universal function approximators (Funahashi & Nakamura, 1993) as long as the activation function is nonlinear. We would like to draw attention to a new experiment we have included with GRUs thanks to the reviewer's suggestion, to highlight the universality of our results. We found that our results do hold even with continuous-time GRUs (Suppl. Fig. A.4), showing that these reactivation phenomena are not unique to a particular network architecture or activation function. We add a discussion about this important point in the revised manuscript.
>
> Funahashi & Nakamura. Approximation of dynamical systems by continuous time recurrent neural networks. Neural Networks 6 (1993): 801-806.
>
> > Regarding the quiescent state, I think the authors are perhaps a little too cavalier with the fact that, strictly speaking, the noise term must change in order to guarantee replay. As presented at the moment, it feels to this reader like a weakness that is not adequately addressed. The authors should either supply 1) a reason to believe this sort of change happens in real networks or 2) a demonstration that the deviations are not sufficient to qualitatively change the main result. There is also the possibility that this is an opportunity for the authors. The deviations from perfect replay if the noise does not change amount to a prediction, and one which may have computational advantages (e.g. facilitating exploration in terms of a planning algorithm).
>
> **Response:** We apologize if there has been a misunderstanding, but the increase in noise variance is **not required** to see reactivation. It is only required for an **exact** distributional match for the Langevin sampling. We demonstrate in Suppl. Fig. A.1 (in the updated manuscript) that reactivation occurs and matches the waking behaviour even when the noise variance remains the same during quiescence. The noise variance merely behaves like a temperature parameter for the sampling, reactivation is not precluded if it stays at the same level. As the reviewer points out, there may indeed be related computational advantages which future work could explore. We thank the reviewer for raising this point and the suggestion for the additional demonstration, which we believe improves our work.

---

> > ### Author Response · Authors · 2023-11-17
> > **Response to Reviewer aZpo**
> >
> > > Last sentence of the introduction: I think "ecologically" should be "ethnologically"
> >
> > **Response:** We thank the reviewer for the suggestion, we have modified this sentence to use "ethologically" instead of "ecologically" in the updated draft.
> >
> > > Does this work for more general RNNs (more general nonlinearities, interactions between recurrence and signal integration, etc.)?
> >
> > **Response:** We do not believe that there is any principled reason why our analyses wouldn't extend to other RNNs. We thank the reviewer for this suggestion, and we decided to check this experimentally with continuous-time GRUs. We found that our results do hold (Suppl. Fig. A.4), showing that these reactivation phenomena are not unique to a particular network architecture or activation function. While our theory does not preclude them (Funahashi & Nakamura, 1993; Maheswaranathan et al., 2019), in practice, GRUs were more sensitive to increases in quiescent noise, and other activation functions would require more hyperparameter tuning for the network's optimal performance.
> >
> > Funahashi & Nakamura. Approximation of dynamical systems by continuous time recurrent neural networks. Neural Networks 6 (1993): 801-806.
> >
> > Maheswaranathan et al. Universality and individuality in neural dynamics across large populations of recurrent networks. Advances in neural information processing systems 2019 (2019): 15629-15641.
> >
> > > Is there reason to believe the noise distribution changes in the quiescent state?
> >
> > **Response:** To our knowledge, there is no evidence that this must occur. However, we would like to reiterate that our main results hold even without the change in noise variance, as stated previously (Suppl. Fig. A.4). Noise only acts like a temperature parameter on the distribution.
> >
> > We believe that we have addressed all of the reviewer's concerns, and would be grateful if the reviewer would consider raising their score. If the reviewer has any additional questions, we would be happy to discuss further. We eagerly await the reviewer's response.

---

> ### Author Response · Authors · 2023-11-20
> **Following up with the reviewer**
>
> Dear Reviewer aZpo,
>
> We believe our response has addressed all of the reviewers' concerns and would be grateful if the reviewer would consider raising their score. As the discussion period is coming to an end soon, we would greatly appreciate a response from the reviewer. If the reviewer has any further questions, we would be happy to address them. We thank the reviewer again for their helpful suggestions and eagerly look forward to a response.

---

> ### Author Response · Authors · 2023-11-22
> **Discussion period ending soon**
>
> Dear Reviewer aZpo,
>
> This is a gentle reminder that the discussion period with authors is coming to an end (<20 hours). We believe our responses in the comments and updated manuscript have addressed all of the reviewers' concerns, and we would be grateful if the reviewer would consider raising their score. If the reviewer has any further questions, we would be happy to address them in the short time remaining. We thank the reviewer again for their helpful suggestions and eagerly look forward to a response.

---

> > ### Comment · Reviewer_aZpo · 2023-11-23
> >
> > Thank you for the thoughtful responses.  I don't think the authors have addressed my primary concerns, which involve the connection between their empirical training and the analytic bound they consider.  I don't see any arguments that provide reasonable guarantees that the two cost functions under consideration will be approximations of each other without additional assumptions.
> >
> > I also understood that changing the noise variance was not required, but this issue is dealt with rather cavalierly in the paper itself and would lead a reasonable reader to think something had been brushed under the rug.  The paper would be strengthened with some more discussion (or examples) of what happens for other choices of noise variance.  The new figure is helpful in this regard.
> >
> > The paper is quite nice and I remain overall of positive opinion about it.

---

### Official Review · Reviewer_pnLC · 2023-10-31

**Soundness:** 2 fair
**Presentation:** 3 good
**Contribution:** 3 good
**Rating:** 6
**Confidence:** 3

**Summary:**

The authors develop a theoretical framework for understand why offline reactivation occurs in recurrent neural networks. In particular, the authors argue that neural noise during awake states is essential for the emergence of faithful and varied reactivation of neural trajectories during quiscience (i.e. in absence of external stimuli).

**Strengths:**

- the paper is generally well written
- the mathematical theory is well presented and interesting (as far as I'm aware it is novel, but I am not certain of this)
- offline reactivation should be of interest both to machine learning and neuroscience researchers; the paper does make some theoretical and experimental headway into why/how it occurs

**Weaknesses:**

- perhaps this is not in the scope of the paper, but the authors do not provide any theoretical/numerical support for the functional benefit of offline reactivation. The authors mention other works which demonstrate that it may aid the formation of long-term memories, schema, planning. Do the theoretical/numerical results in this paper support these possibilities, or any one in particular?
- Relatedly, a key feature of this study is that the RNNs considered are noisy during awake states. It's unclear to me exactly why this is necessary. The authors "found that even trajectories generated by networks that were not trained in the presence of noise, and also were not driven by noise in the quiescent phase, still generated quiescent activity distributions that corresponded well to the active phase distributions"; is the point that there's more space exploration with noise, giving rise to a functional benefit? Or that somehow the introduction of noise better captures observed neural data?
- I found the link between the theoretically optimal solution for update dynamics and the actual application of RNNs somewhat unclear. Are we to take that (presumably backprop-trained) RNNs employ update dynamics similarly to the theoretical solution; is it biologically reasonable for e.g. equation 18 to be implement by an RNN? If so, is it possible to show that they do; e.g. by comparing these recurrent inputs to the optimal solution? If not, are the experimental results relevant to the theory?
- Relatedly, perhaps this is overly harsh, but I also do not find the result that RNNs without stimuli visit similar states as with stimuli very surprising.
- The authors suggest an interesting discrepancy in experimental prediction for their noise-based theory of offline reactivation and generative modeling; specifically, that "while generative models necessarily recapitulate the moment-to-moment transition statistics of sensory data, our approach only predicts that the stationary distribution will be identical". I am confused by this statement and not sure if it is true. Is that to say that the proposed model would not encode sensory transitions but would simply replicate the overall probability distribution of sensory states? I would find this surprising for an RNN whose recurrent weights are trained on sensory data; that is, I would suppose that the RNN weights would themselves somehow capture transition statistics of the task variables (e.g. maybe this is valuable as a kind of denoising effect on noisy observations). Perhaps I am wrong though

**Questions:**

- In section 2.1 it may be relatively obvious that p denotes probability, but I would still clarify this
- In equation 15 it's a bit odd to provide a new definition for L_{noise} having just defined it previousply with the D term involved
- the delta t is given as 0.02. What's the unit, seconds?
- I would recommend a limitations section

---

> ### Author Response · Authors · 2023-11-17
> **Response to Reviewer pnLC**
>
> We thank the reviewer for their detailed review. We appreciate that the reviewer found our paper to be "well written" and the theory "well presented and interesting". We have attempted to address the reviewer's concerns and questions below:
>
> > perhaps this is not in the scope of the paper, but the authors do not provide any theoretical/numerical support for the functional benefit of offline reactivation. [...] Do the theoretical/numerical results in this paper support these possibilities, or any one in particular?
>
> **Response:** This is indeed an interesting research question. However, as the reviewer themselves point out, we believe this is out of scope for our current work---our scope here is to explain theoretically the emergence of reactivation and justify this with experiments, not to identify its functional benefits. It is nevertheless an important direction for future work. Many papers have talked about the functional benefits of offline reactivation in other contexts (Mnih et al. 2015; Hayes et al. 2019; Hayes & Kanan 2022), but the reviewer is correct in noting that we would have to concretely demonstrate the benefits of this particular form of reactivation. We acknowledge that this is an immediate direction to be explored and are revising our discussion to discuss the implications of our results in this context.
>
> Mnih et al. Human-level control through deep reinforcement learning. Nature 518, 529–533, 2015.
>
> Hayes et al. Memory efficient experience replay for streaming learning. International Conference on Robotics and Automation (ICRA), pp. 9769–9776. IEEE, 2019.
>
> Hayes & Kanan. Online continual learning for embedded devices. arXiv preprint arXiv:2203.10681, 2022.
>
> > Relatedly, a key feature of this study is that the RNNs considered are noisy during awake states. It's unclear to me exactly why this is necessary. The authors "found that even trajectories generated by networks that were not trained in the presence of noise, and also were not driven by noise in the quiescent phase, still generated quiescent activity distributions that corresponded well to the active phase distributions"; is the point that there's more space exploration with noise, giving rise to a functional benefit? Or that somehow the introduction of noise better captures observed neural data?
>
> **Response:** Indeed, as the reviewer points out, the noise leads to more exploration. Noise has been shown to improve RNN training (Qin & Vučinić, 2018). Furthermore, neural activity in the brain is noisy (Destexhe & Rudolph-Lilith, 2012), for example, in the probabilistic nature of synaptic transmission, and in other ways.
>
> It is critical to note that the match in distributions is across multiple trajectories (multiple initializations), and so does not entirely capture the properties of individual trajectories, such as the amount of exploration.
>
> We thus carried out an experiment to show the amount of exploration of these noiseless trajectories. Indeed we found that individual trajectories do not explore the space well, and get stuck (Suppl. Fig. A.2c-d). Adding noise during quiescence alone does not fix this problem---it makes the trajectories erratic and implausible (Suppl. Fig. A.2e-f), with much higher point-to-point distances than even waking behavior (Suppl. Fig. A.3). So individual trajectories from noiseless networks do a worse job relative to the noisy trained networks. Noisy training stabilizes noisy quiescent activity, which in turn explores more of the task manifold than noiseless quiescent activity. We clarify this point in the revised manuscript.
>
> Qin & Vučinić. Training Recurrent Neural Networks against Noisy Computations during Inference. 2018 52nd Asilomar Conference on Signals, Systems, and Computers (2018): 71-75.
>
> Destexhe & Rudolph-Lilith. Neuronal Noise. Springer, 2012.

---

> ### Author Response · Authors · 2023-11-17
> **Response to Reviewer pnLC**
>
> > I found the link between the theoretically optimal solution for update dynamics and the actual application of RNNs somewhat unclear. Are we to take that (presumably backprop-trained) RNNs employ update dynamics similarly to the theoretical solution; is it biologically reasonable for e.g. equation 18 to be implement by an RNN? If so, is it possible to show that they do; e.g. by comparing these recurrent inputs to the optimal solution? If not, are the experimental results relevant to the theory?
>
> **Response:** We apologize if this was not clear. The idea is that the RNNs trained optimally (following any optimization scheme) are approximating the ground-truth optimal solution, given by the greedily optimal dynamics (Section 2.3). It is indeed reasonable for an RNN to implement Eq. 18---according to the literature on RNNs being universal function approximators (Funahashi & Nakamura, 1993). We did not show that the network dynamics explicitly align with the optimal solution, though our empirical results indicate that they do. Indeed we see the Langevin sampling during quiescence leading to the distributional match. It is certainly possible for suboptimal (or very small) RNNs to fail to align with the theory, the theory only applies for very well-trained networks that are performing near-optimally such as our networks. We clarify these points in the revised manuscript.
>
> Funahashi & Nakamura. Approximation of dynamical systems by continuous time recurrent neural networks. Neural Networks 6 (1993): 801-806.
>
> > Relatedly, perhaps this is overly harsh, but I also do not find the result that RNNs without stimuli visit similar states as with stimuli very surprising.
>
> **Response:** We believe that there may have been a misunderstanding---our claim is much stronger than merely saying that similar states are visited in the absence of stimuli. Instead, we prove and demonstrate that the networks visit the same states with the same, i.e., **identical steady-state distribution** during quiescence as active behavior. During our biased and unbiased tasks, the same path integrating continuous attractor could theoretically solve both tasks, but then rehearsal would be the same in both conditions. We see that the networks learn different solutions for the biased and unbiased tasks, and that these different solutions produce rehearsals that reflect the task statistics. This is non-trivial, and demonstrating this is, we believe, our main novel contribution. We further clarify this point in the revised manuscript.
>
> > The authors suggest an interesting discrepancy in experimental prediction for their noise-based theory of offline reactivation and generative modeling; specifically, that "while generative models necessarily recapitulate the moment-to-moment transition statistics of sensory data, our approach only predicts that the stationary distribution will be identical". I am confused by this statement and not sure if it is true. Is that to say that the proposed model would not encode sensory transitions but would simply replicate the overall probability distribution of sensory states? I would find this surprising for an RNN whose recurrent weights are trained on sensory data; that is, I would suppose that the RNN weights would themselves somehow capture transition statistics of the task variables (e.g. maybe this is valuable as a kind of denoising effect on noisy observations). Perhaps I am wrong though
>
> **Response:** The reviewer is indeed correct that it is **possible** for our networks to capture the transition dynamics (and this is what happens in practice), it is just that our theory does not comment on the dynamics of reactivations. In a new experiment with training solely on counter-clockwise trajectories for the head direction task (Suppl. Fig. A.5), we found that the reactivations recapitulated the transition structure of active phase trajectories, and maintained their bearing for longer periods, resembling real neural activity more closely. This would indeed appear to be due to the RNN weights capturing the transition structure, but is not a prediction of our theory. The biased velocities were reflected during quiescence, but less clearly than during waking. However, reversals in the trajectories still occurred (Suppl. Fig. A.5b).

---

> > ### Author Response · Authors · 2023-11-17
> > **Response to Reviewer pnLC**
> >
> > > In section 2.1 it may be relatively obvious that p denotes probability, but I would still clarify this
> >
> > **Response:** We thank the reviewer for the suggestion, we have updated this in the draft.
> >
> > > In equation 15 it's a bit odd to provide a new definition for L_{noise} having just defined it previously with the D term involved
> >
> > **Response:** We have used the `\equiv` ($\equiv$) symbol to denote an equivalence between the location of minima for $\mathcal{L}_{noise}$ (which was defined previously) and the RHS of Eq. 15. We apologize if this was not clear. We have updated the text above to clearly state the equivalence, and modified the equation to be more explicit about the equivalence in location of minima.
> >
> > > the delta t is given as 0.02. What's the unit, seconds?
> >
> > **Response:** Yes, the unit is indeed seconds. We have updated the draft to add the missing unit, and thank the reviewer for the observation.
> >
> > > I would recommend a limitations section
> >
> > **Response:** We have updated the Discussion in the draft to mention the limitations of our work. We have also included a more detailed breakdown of the assumptions (and hence limitations) of our work in the Supplementary Material.
> >
> > We believe that we have addressed all of the reviewer's concerns, and would be grateful if the reviewer would consider raising their score. If the reviewer has any additional questions, we would be happy to discuss further. We eagerly await the reviewer's response.

---

> > > ### Comment · Reviewer_pnLC · 2023-11-19
> > >
> > > Thank you to the authors for their detailed response and clarifications. I also appreciate the new supplementary figures and analyses.
> > >
> > > Overall, I do believe the mathematical analysis is interesting enough for publication, but I would encourage the authors to better link their theory with backprop-trained RNNs and try to present (if only preliminary) emperical results which demonstrate the functional utility of their theory.
> > >
> > > I will modify my score accordingly.

---

> > > > ### Author Response · Authors · 2023-11-20
> > > > **Response to Reviewer pnLC**
> > > >
> > > > We thank the reviewer for their timely response, and are grateful that the reviewer appreciates our responses and new analyses. We are encouraged that the reviewer finds our mathematical analysis interesting, and thank the reviewer for raising their score.
> > > >
> > > > We believe that our current work is self-contained and complete, and focuses on **explaining the emergence of reactivation** in neural circuits. While there are extensions to our work that could explore the utility of reactivations, in our opinion these would have to explored as separate future work as they would require a lot of effort, and would be difficult to accomplish in the short time available.
> > > >
> > > > For example, we could explore using rehearsal activity as an input to a separate memory network that is forming neuronal assemblies through synaptic plasticity (e.g. a Hopfield network). The extra input data could facilitate consolidation. This is in line with the complementary learning systems idea (McLelland et al., 1995). We could also explore how this rehearsal mechanism interacts with continual learning algorithms (either synaptic consolidation mechanisms from neuroscience or ML methods like elastic weight consolidation).
> > > >
> > > > However, exploring these would require highly non-trivial modifications to our setup, which we cannot achieve in the time available. We thank the reviewer again for their helpful review and response.
> > > >
> > > > McClelland et al. Why there are complementary learning systems in the hippocampus and neocortex: Insights from the successes and failures of connectionist models of learning and memory. Psychological Review, 102(3), 419–457. 1995.

---

### Official Review · Reviewer_xvHM · 2023-10-31

**Soundness:** 2 fair
**Presentation:** 2 fair
**Contribution:** 2 fair
**Rating:** 5
**Confidence:** 4

**Summary:**

This submission seeks to model the reactivation of brain activity during periods of quiescence using task-optimized network. The paper develops some mathematical formalisms to show that under certain assumptions, recurrent neural networks trained to perform a task develop denoising dynamics.

While there are some interesting ideas presented in this submission, overall it feels that the results presented in the current version are preliminary and not surprising.

**Strengths:**

— Understanding the neural mechanisms underlying reactions/replays is an important question.

-- The paper seeks to analyze RNNs trained to perform a class of tasks theoretically, which is a somewhat rare excise in this literature.

— The mathematical analysis of the loss function used in the training and the connection to the Langevin dynamics, while under somewhat strong assumptions, remain interesting.

**Weaknesses:**

1. A number of relevant studied were not cited and discussed. This include work using RNNs and attractor dynamics to model the replay and theta sequences in the hippocampus, e.g.,
Hopfield, John J. "Neurodynamics of mental exploration." Proceedings of the National Academy of Sciences 107.4 (2010): 1648-1653.
Kang, Louis, and Michael R. DeWeese. "Replay as wavefronts and theta sequences as bump oscillations in a grid cell attractor network." Elife 8 (2019): e46351.
Chu, Tianhao, et al. "Firing rate adaptation affords place cell theta sweeps, phase precession and procession." bioRxiv (2022): 2022-11.

The paper also misses several pieces work in training RNNs to study the grid cells systems and HD systems.
Cueva, C. J., & Wei, X. X. (2018). Emergence of grid-like representations by training recurrent neural networks to perform spatial localization. ICLR.
Uria, B., Ibarz, B., Banino, A., Zambaldi, V., Kumaran, D., Hassabis, D., Barry, C. and Blundell, C., (2020). The spatial memory pipeline: a model of egocentric to allocentric understanding in mammalian brains. BioRxiv, pp.2020-11.
Cueva, C.J., et al (2020). Emergence of functional and structural properties of the head direction system by optimization of recurrent neural networks.ICLR.

2. The main results are not surprising after considering what we know so far on this topic. Prior work has shown that training the RNNs to perform path integration or angular integration task leads to networks that exhibit attractors dynamics that is similar to continuous attractor models. Furthermore, it is well known that attractor dynamics can perform denoising. Thus it is not surprising that RNNs trained on these previous studied tasks can perform denoising.

It should also be noted that prior work has further characterize the rate of diffusion along the low-d manifold, e.g., see Fig 6 of the following paper:
Burak, Y. and Fiete, I.R., 2009. Accurate path integration in continuous attractor network models of grid cells. PLoS computational biology, 5(2), p.e1000291.


3. As described in the paper, the trained network follows diffusion dynamics when the stimulus turned off. This is naturally expected because the diffusion of network state on a low-d manifold. This would result in brownian-motion like trajectory. Furthemore, the replay trajectories observed in the hippocampus typically described by a systematic drift towards one direction (e.g., in linear or circular tracks), not diffusive dynamics. Can the authors be more explicit about the experimental data they were modeling?

4. One more point that confuses me—the authors stated “we found that even trajectories generated by networks that were not trained in the presence of noise, and also were not driven by noise in the quiescent phase, still generated quiescent activity distributions that corresponded well to the active phase distributions. ” This seems to argue against the denoising dynamics, and make the usefulness of the mathematical analysis questionable.

5. The mathematical analysis relied on a set of approximations and assumptions, which were not well justified.

**Questions:**

Would it be possible to discuss more explicitly how their results are connected or supported by the empirical data? In particular, how is the “diffusive reactivation” related to neural data?

 The mathematical analysis relies on a number of assumptions, to the extent that it is difficult to judge whether the conclusion would actually be applicable to the numerical experiments. Can these assumptions be justified or better motivated (beyond the sake of mathematical convenience)?

---

> ### Author Response · Authors · 2023-11-17
> **Response to Reviewer xvHM**
>
> We thank the reviewer for reviewing our paper and providing helpful comments that we believe lead us to considerably enrich our work. We are glad that the reviewer found our theoretical approach, which the reviewer has stated is a rare exercise in the literature, to be interesting and addressing an important question. We have provided our responses to the reviewer's comments below:
>
> > A number of relevant studied were not cited and discussed. This include work using RNNs and attractor dynamics to model the replay and theta sequences in the hippocampus [...] The paper also misses several pieces work in training RNNs to study the grid cells systems and HD systems. [...]
>
> **Response:** We thank the reviewer for pointing us to these works. We have cited these in our updated draft. We would also like to stress that the focus of our work is to demonstrate that reactivations are statistical reproductions of waking activity (i.e. the steady-state distribution of reactivations is **identical** to that of waking activity), and not just observed as a result of our networks learning continuous attractors. We formalize and prove this mathematically, and also show this with experiments. We apologize if this was not clear. We hope that our comments here and in the updated draft improve on this.
>
> > The main results are not surprising after considering what we know so far on this topic. Prior work has shown that training the RNNs to perform path integration or angular integration task leads to networks that exhibit attractors dynamics that is similar to continuous attractor models. Furthermore, it is well known that attractor dynamics can perform denoising. Thus it is not surprising that RNNs trained on these previous studied tasks can perform denoising.
>
> **Response:** We apologize if the main message appeared to be the related to the RNNs learning continuous attractors and performing denoising, this is not our intention. In fact, our very starting point is the recognition that continuous attractor networks emerge with training, it is not one of our major claims. The central finding and novel contribution of our manuscript is to explore the conditions under which reactivation, driven by noise, matches the statistics of waking activity.
>
> With respect to denoising, we believe that one of our novel contributions is to show theoretically how denoising comes to be, and gives rise to statistically faithful reactivations as described below, providing a principled theoretical understanding. We would like to reiterate that our focus is not to show that denoising occurs, although we build on works that explore this idea (Section 2.2--2.3).
>
> More precisely, our novel contribution is to prove and demonstrate that the steady-state distribution of reactivations is **identical** to that of waking activity, under a well-defined and minimal set of conditions. While this mechanism is related to denoising, which will occur along attractor manifolds, our contribution is to demonstrate that optimal denoising under different task statistics will show rehearsal patterns that reflect those different task statistics. This is the key novel contribution, which we believe is non-trivial in and of itself. We make sure to clearly communicate our findings in the revised manuscript (Section 2.4).

---

> > ### Author Response · Authors · 2023-11-17
> > **Response to Reviewer xvHM**
> >
> > > As described in the paper, the trained network follows diffusion dynamics when the stimulus turned off. This is naturally expected because the diffusion of network state on a low-d manifold. This would result in brownian-motion like trajectory. Furthermore, the replay trajectories observed in the hippocampus typically described by a systematic drift towards one direction (e.g., in linear or circular tracks), not diffusive dynamics. Can the authors be more explicit about the experimental data they were modeling?
> >
> > and
> >
> > > Would it be possible to discuss more explicitly how their results are connected or supported by the empirical data? In particular, how is the “diffusive reactivation” related to neural data?
> >
> > **Response:** We apologize if this was not clear, but as stated previously, the focus of our mathematical analysis is on the resulting steady-state distributions of quiescent activity, rather than the transient moment-to-moment transition statistics. While diffusion-like motion on the attractor manifold has been described previously, to our knowledge it has not been formally or theoretically explored how this gives rise to statistically faithful reactivations. Moreover, it is not clear whether this reactivation phenomenon can be well-described by a formal diffusion process, and what are the conditions under which autonomous network dynamics would lead to such activity.
> >
> > Although experts might find the presence of reactivation intuitive, our work formalizes this phenomenon and provides critical details about the statistics of reactivation distributions. We kindly note that our work gains importance specifically by formalizing an observation that seems intuitive, as it now provides a theoretical grounding to study reactivation.
> >
> > While we have not modeled any specific experimental data for circular and linear tracks, our results do not state anything specific about the sequential dynamics on the manifold, only that the steady-state distribution is matched to waking statistics. However, in a new experiment with the head direction task (Suppl. Fig. A.5), we biased the waking moment-to-moment transition structure to be counter-clockwise. We found that quiescent trajectories partially recapitulated the transition structure of active phase trajectories, and maintained their bearing for longer periods, resembling real neural activity more closely. The biased velocities were also reflected during quiescence, although less clearly than waking. However, reversals in the trajectories still occurred.
> >
> > > One more point that confuses me—the authors stated “we found that even trajectories generated by networks that were not trained in the presence of noise, and also were not driven by noise in the quiescent phase, still generated quiescent activity distributions that corresponded well to the active phase distributions. ” This seems to argue against the denoising dynamics, and make the usefulness of the mathematical analysis questionable.
> >
> > **Response:** While we do observe this, it is critical to note that the match in distributions is across multiple trajectories (multiple initializations), and so does not entirely capture the properties of individual trajectories, such as the amount of exploration.
> >
> > We thus carried out an experiment to show the amount of exploration of these noiseless trajectories. Indeed we found that individual trajectories do not explore the space well, and get stuck (Suppl. Fig. A.2c-d). Adding noise during quiescence alone does not fix this problem---it makes the trajectories erratic and implausible (Suppl. Fig. A.2e-f), with much higher point-to-point distances than even waking behavior (Suppl. Fig. A.3). So individual trajectories from noiseless networks do a worse job relative to the noisy trained networks. Noisy training stabilizes noisy quiescent activity, which in turn explores more of the task manifold than noiseless quiescent activity.

---

> ### Author Response · Authors · 2023-11-17
> **Response to Reviewer xvHM**
>
> > The mathematical analysis relied on a set of approximations and assumptions, which were not well justified.
>
> and
>
> > The mathematical analysis relies on a number of assumptions, to the extent that it is difficult to judge whether the conclusion would actually be applicable to the numerical experiments. Can these assumptions be justified or better motivated (beyond the sake of mathematical convenience)?
>
> **Response:** We thank the reviewer for the suggestion, and we have added a section to the Supplementary Material to clarify the key assumptions for our theory. We provide the assumptions and explanations here:
> 1. We consider discrete-time approximations of noisy continuous-time RNNs. Using continuous-time RNNs is a common practice in the literature, and we use its discrete-time approximation to be in line with our implementation.
> 2. The network must be performing some variant of path integration, by integrating change-based information about environmental state variables to some function of these variables. This condition is often met in natural settings, such as spatial navigation.
> 3. We assume that the inputs to the network are drawn from a stationary distribution, which would translate to assuming that the effects of initial conditions on state occupancy statistics are ignored, and also assuming that the behavioral policy remains constant throughout time.
> 4. We consider greedy optimization of the loss at every timestep. Greedy optimization is a sensible way of partitioning effort across time in this task: the agent does the best that it can on each timestep, assuming that at each previous timestep the best possible job has been done. In the absence of noise the greedily optimal solution is equivalent to path integration, which is also a **globally** optimal solution.
> 5. We assume that the network is performing optimally in the presence of noise. In practice, we train our networks until their loss reaches very low, near-zero values.
> 6. We assume that our network dynamics can be decomposed into two terms with different functional dependencies. The first term depends only on the activity while the second term depends on all original dependencies—the activity, the state and the change-based inputs.
> 7. For the quiescent state, we assume that the change-based sensory inputs are zero. This is reasonable because for tasks like those we have considered, which involve integrating self-motion cues, $\frac{\mathrm{d}s(t)}{\mathrm{d}t}$ must be zero during periods of quiescence like sleep, where the animal is not moving and hence doesn't receive sensory inputs associated with self-motion.
> 8. While we assume that the noise variance is doubled during quiescence to show exact equivalence with Langevin sampling, this is by no means necessary to witness reactivation (Suppl. Fig. A.4). This noise variance is equivalent to a temperature parameter for the sampling.
>
> RNNs trained optimally (following any optimization scheme) are approximating the ground-truth optimal solution, given by the greedily optimal dynamics (Section 2.3). It is indeed reasonable for an RNN to implement Eq. 18---according to the literature on RNNs being universal function approximators (Funahashi & Nakamura, 1993). Our empirical results indicate that the network's dynamics align with this solution. Indeed we see the Langevin sampling during quiescence leading to the distributional match. It is certainly possible for suboptimal (or very small) RNNs to fail to align with the theory, the theory only applies for very well-trained networks that are performing near-optimally such as our networks. Overall, we believe our assumptions are well-justified, and the conclusions of the theory are indeed applicable to our numerical experiments as evidenced by our results.
>
> Funahashi & Nakamura. Approximation of dynamical systems by continuous time recurrent neural networks. Neural Networks 6 (1993): 801-806.
>
> We hope that our response and updates to the draft have addressed all of the reviewer's concerns, and would be grateful if the reviewer would consider raising their score. If the reviewer has any additional questions, we would be happy to discuss further. We eagerly await the reviewer's response.

---

> > ### Author Response · Authors · 2023-11-20
> > **Following up with the reviewer**
> >
> > Dear Reviewer xvHM,
> >
> > We believe our response has addressed all of the reviewers' concerns and would be grateful if the reviewer would consider raising their score. As the discussion period is coming to an end soon, we would greatly appreciate a response from the reviewer.  If the reviewer has any further questions, we would be happy to address them. We thank the reviewer again for their helpful suggestions and eagerly look forward to a response.

---

> ### Comment · Reviewer_xvHM · 2023-11-22
> **thanks for the clarifications**
>
> I appreciate the authors’ detailed response to my critiques. The response helps clarify some of the concerns. In particular, the key contribution is made more clear.  I have raised my score.
>
> A couple of issues:
> -- while the relevant references were added to the revised version, generally there is a lack of appropriate discussion of these studies in the context.
>
> -- the connection to empirical research/ neuroscience data remains to be strengthened. I don’t feel this point was appropriately addressed in the revised version.

---

> > ### Author Response · Authors · 2023-11-22
> > **Response to Reviewer xvHM**
> >
> > We thank the reviewer for going through our responses and revised manuscript. We appreciate the reviewer's detailed and helpful suggestions that have helped us considerably improve our work. We apologize that our discussions of some relevant work and connections to neuroscience data were incomplete. Thanks to the reviewer's suggestions, we have uploaded a new revision with some key changes:
> >
> > > while the relevant references were added to the revised version, generally there is a lack of appropriate discussion of these studies in the context.
> >
> > We have discussed the relevant works suggested by the reviewer in greater detail, in the context of our work. Here are specific changes we have made to improve our discussion (italicized portions indicate key changes):
> >
> > * **Introduction, paragraph 3: discussing previous 'emergent' modeling approaches**: [...] local synaptic plasticity mechanism (Litwin-Kumar & Doiron, 2014; Theodoni et al., 2018; Haga & Fukai, 2018; Asabuki & Fukai, 2023; _Hopfield, 2010). Other approaches have modeled replay and theta sequences in the hippocampus as emergent consequences of firing rate adaptation (Chu et al., 2023) or input modulation (Kang & DeWeese, 2019) in continuous attractor network models._ [...]
> >
> > * **Introduction, paragraph 4: discussing works training RNNs to study grid/HD cells**: We trained continuous-time recurrent neural networks (RNNs) to optimally integrate and track perceptual variables based on sensations of change (angular velocity, motion through space, etc.), in the context of two ethologically relevant tasks: spatial navigation and head direction integration. _Critically, training in this manner has been shown to produce grid cell (Cueva & Wei, 2018; Sorscher et al., 2019) and head direction cell representations (Cueva et al., 2020; Uria et al., 2020), which correspond to neural systems in which reactivation phenomena have been observed (Gardner et al., 2022; Peyrache et al., 2015)._ We see that these networks exhibit reactivation during quiescent states [...]
> >
> > * **Discussion, paragraph 2:** Previous models have been able to reproduce these reactivation dynamics, _by embedding a smooth attractor in the network’s recurrent connectivity along which activity may diffuse during quiescence (Burak & Fiete, 2009; Khona & Fiete, 2022)._ Similarly, we identified attractors in our trained networks’ latent activation space [...] beyond previous studies, we were able to show that the statistics of reactivation in our trained networks mimicked the statistics of activity during waking behavior, and that manipulation of waking behavioral statistics was directly reflected in offline reactivation dynamics (Fig. 2). Thus, our work complements these previous studies by providing a mathematical justification for the emergence of reactivation dynamics in terms of optimal task performance.
> >
> > > the connection to empirical research/ neuroscience data remains to be strengthened. I don’t feel this point was appropriately addressed in the revised version.
> >
> > We thank the reviewer for this suggestion, and we have updated our manuscript to discuss this in greater detail. In particular, we draw attention to the following:
> >
> > * **Numerical Experiments, Head Direction Estimation (last paragraph): discussing connections to experimental data:** To explore whether reactivation dynamics also recapitulate the moment-to-moment transition structure of waking activity [...] quiescent trajectories partially recapitulated the transition structure of active phase trajectories, and maintained their bearing for longer periods, _resembling real neural activity more closely. The biased velocities were reflected during quiescence, but less clearly than during waking. However, reversals in the trajectories still occurred. These results demonstrate that the type of ‘diffusive’ rehearsal dynamics explored by our theory are still able to produce the temporally correlated, sequential reactivation dynamics observed in the head direction system (Peyrache et al., 2015)._
> >
> > * In the **discussion**, we have discussed our work in the context of:
> >   1. **2nd paragraph:** explaining reactivations in brain regions that perform path integration for navigation and head direction estimation
> >   2. **4th paragraph:** explaining reactivation in various other systems with continuous attractor dynamics which have been observed, in a diverse set of experimental studies.
> >
> > We hope that our updated manuscript adequately addresses the reviewer's concerns, and we would be grateful if the reviewer could consider raising their score. We would like to thank the reviewer again for their thorough review and helpful suggestions to improve our work. We eagerly look forward to the reviewer's response.

---

### Author Response · Authors · 2023-11-20
**General comment on key clarifications and additional results**

We thank all the reviewers for their helpful reviews and suggestions. We are grateful that reviewers in general found our work interesting, and also well-written and well-supported. We believe that the reviewers' comments have helped us considerably improve our work. In this comment, we would like to clarify some key points brought up by reviewers and also discuss additional experiments we have included.

### **Main result**

The central finding and novel contribution of our manuscript is to explore the conditions under which reactivation, driven by noise, matches the statistics of waking activity. More precisely, our novel contribution is to prove and demonstrate that the **steady-state distribution of reactivations is identical to that of waking activity**, under a well-defined and minimal set of conditions. This is a stronger claim than stating that RNNs visit the same states during quiescence as they did during waking.

Furthermore, while this mechanism is related to denoising, which has been observed to occur along attractor manifolds, our contribution is to demonstrate that **optimal denoising under different task statistics will show rehearsal patterns that reflect those different task statistics**. This is the key novel contribution, which we believe is non-trivial in and of itself. We make sure to clearly communicate our findings in the revised manuscript (Section 2.4).

### **Importance of noise during training**

The key observation is that **noisy training improves exploration in quiescent trajectories**. Noise has been shown to improve RNN training (Qin & Vučinić, 2018). Furthermore, neural activity in the brain is noisy (Destexhe & Rudolph-Lilith, 2012), for example, in the probabilistic nature of synaptic transmission, and in other ways.

While noiseless networks also have similar distributions of waking and quiescent activity, it is critical to note that the match in distributions is across multiple trajectories (multiple initializations). This does not entirely capture the properties of individual trajectories, such as the amount of exploration.

We thus carried out an experiment to show the amount of exploration of these noiseless trajectories. Indeed we found that individual trajectories do not explore the space well, and get stuck (Suppl. Fig. A.2c-d). Adding noise during quiescence alone does not fix this problem--it makes the trajectories erratic and implausible (Suppl. Fig. A.2e-f), violating the actual map that was formed. In a new experiment, we quantified this distributionally and showed that these trajectories have much higher point-to-point distances than even waking behavior, indicating clearly that they are discontinuous and implausible (Suppl. Fig. A.3).

Thus, individual trajectories from noiseless networks do a worse job relative to the noisy trained networks. Noisy training stabilizes noisy quiescent activity, which in turn explores more of the task manifold than noiseless quiescent activity. We clarify this point in the revised manuscript.

Qin & Vučinić. Training Recurrent Neural Networks against Noisy Computations during Inference. 2018 52nd Asilomar Conference on Signals, Systems, and Computers (2018): 71-75.

Destexhe & Rudolph-Lilith. Neuronal Noise. Springer, 2012.

### **Noise variance during quiescence**

The increase in noise variance by a factor of 2 during quiescence is **not required** to see reactivation. It is only required for an **exact** distributional match for the Langevin sampling. We demonstrate in Suppl. Fig. A.1 (in the updated manuscript) that reactivation occurs and matches the waking behaviour even when the noise variance remains the same during quiescence. The noise variance merely behaves like a temperature parameter for the sampling, reactivation is not precluded if it stays at the same level.

### **Capturing transition dynamics**

Although our theory does not constrain quiescent trajectories to match waking trajectories in their transition sequences, it is **possible** for our networks to capture the transition dynamics (and this is what happens in practice). In a new experiment with training solely on counter-clockwise trajectories for the head direction task (Suppl. Fig. A.5), we found that the reactivations recapitulated the transition structure of active phase trajectories, and maintained their bearing for longer periods, resembling real neural activity more closely. The biased velocities were reflected during quiescence, but less clearly than during waking. However, reversals in the trajectories still occurred (Suppl. Fig. A.5b).

---

> ### Author Response · Authors · 2023-11-20
> **General comment on key clarifications and additional results**
>
> ### **Assumptions related to RNNs for theoretical result (and universality of results)**
>
> Our mathematical analysis is a theory for high capacity networks capable of legitimately approximating the optimal solution. The idea is that the RNNs trained optimally are approximating the ground-truth optimal solution, given by the greedily optimal dynamics (Section 2.3). It is indeed reasonable for an RNN to implement Eq. 18--according to the literature on RNNs being universal function approximators (Funahashi & Nakamura, 1993), given that the activation function is nonlinear. We are thus assuming that our trained RNNs are near the optimum, and our empirical results indicate that they are.
>
> Our theoretical results do not make any assumptions about the architecture of the RNN (in fact we have added simulations with GRUs, Suppl. Fig. A.4), only that it is trained to be optimal. Indeed we see the Langevin sampling during quiescence leading to the distributional match. It is certainly possible for suboptimal (or very small) RNNs to fail to align with the theory, the theory only applies for very well-trained networks that are performing near-optimally such as our networks.
>
> We would like to draw attention to a new experiment we have included with GRUs to highlight the universality of our results. We found that our results do hold even with continuous-time GRUs (Suppl. Fig. A.4), showing that these reactivation phenomena are not unique to a particular network architecture or activation function. We add a discussion about this important point in the revised manuscript.
>
> Funahashi & Nakamura. Approximation of dynamical systems by continuous time recurrent neural networks. Neural Networks 6 (1993): 801-806.
>
> Once again, we thank all the reviewers for their suggestions. We believe that we have addressed all reviewers' concerns in our comments, and eagerly look forward to their responses.

---

### Public Comment · ~Nanda_H_Krishna1 · 2025-05-23

We identified a minor issue in Section 2.2, where the noise timesteps are off by $\Delta t$. This has been fixed in the arXiv version (https://arxiv.org/abs/2505.17003) and overall, the theoretical result remains unchanged. We request readers to refer to the arXiv version of the paper henceforth. Apologies for the inconvenience!

---

### Meta-Review · Area_Chair_Vq2d · 2023-12-12

**Metareview:**

The submission demonstrates how recurrent neural network dynamics can separate into denoising and signal integration components, and that this decomposition imposes structure on the distribution of reactivations in the absence of signal input, offering new potential insights into neural replay. The mix of theoretical and numerical analysis is solid, making it a valuable interdisciplinary contribution to the conference.

**Justification For Why Not Higher Score:**

scope of the contributions may not have a broad audience at ICLR

**Justification For Why Not Lower Score:**

solid mixed theory-empirical contribution

---

### Decision · Program_Chairs · 2024-01-16

Accept (poster)